# VIVIDCAM: Learning Unconventional Camera Motions from Virtual Synthetic Videos

Qiucheng Wu [1 2 *]   Handong Zhao [2]   Zhixin Shu [2]   Jing Shi [2]   Yang Zhang [3]   Shiyu Chang [1]

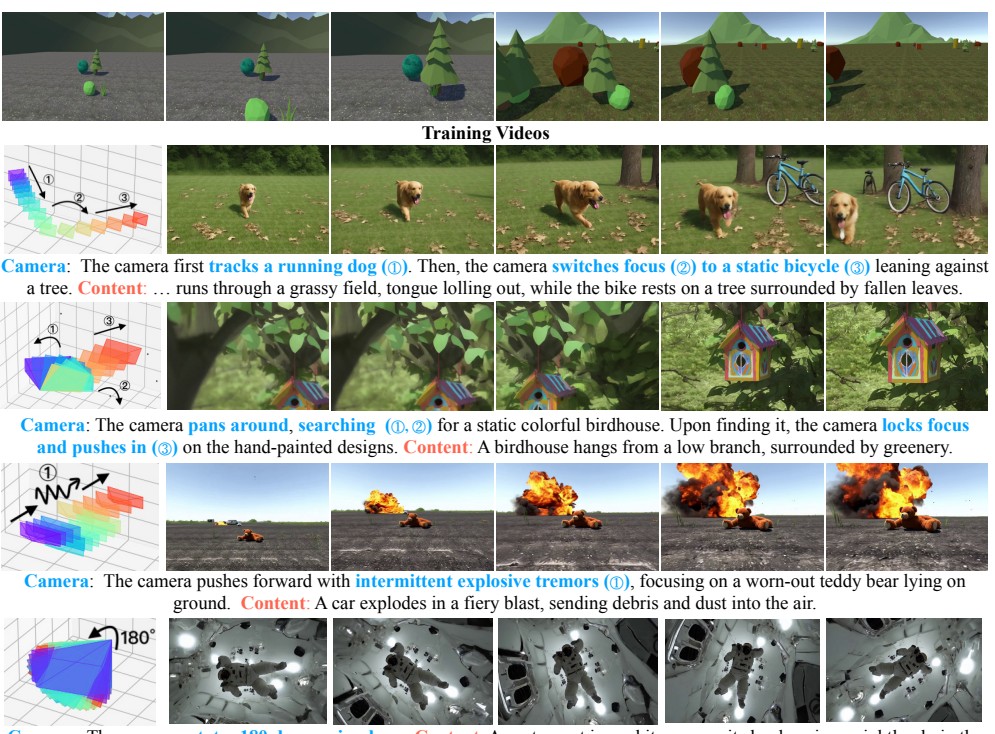

Camera: The camera first **tracks a running dog (①)**. Then, the camera **switches focus (②) to a static bicycle (③)** leaning against a tree. **Content**: … runs through a grassy field, tongue lolling out, while the bike rests on a tree surrounded by fallen leaves.

Camera: The camera **pans around, searching (①, ②)** for a static colorful birdhouse. Upon finding it, the camera **locks focus and pushes in (③)** on the hand-painted designs. **Content**: A birdhouse hangs from a low branch, surrounded by greenery.

Camera: The camera pushes forward with **intermittent explosive tremors (①)**, focusing on a worn-out teddy bear lying on ground. **Content**: A car explodes in a fiery blast, sending debris and dust into the air.

Camera: The camera **rotates 180 degrees in place**. **Content**: An astronaut in a white spacesuit slowly spins weightlessly in the silence of outer space in the spaceship.

*Figure 1.* **VIVIDCAM** learns diverse unconventional camera motions from synthetic videos. The training data (1st row) are simple low-poly 3D scenes rendered in Unity in about *5 seconds* per video. In contrast, the generated results show high visual quality with meaning-driven motions that convey intention (2nd–3rd rows) and more dramatic, unusual motions for artistic effect (4th–5th rows).

## Abstract

Although recent video generative models are getting more capable of following external camera controls, imposed by either text descriptions or camera trajectories, they still struggle to generalize to unconventional camera motions, which is crucial in creating truly original and artistic videos. The challenge lies in finding sufficient training videos with the intended uncommon camera motions. To this end, we propose VIVIDCAM, a training paradigm that enables diffusion models to learn complex camera motions from synthetic videos, releasing the reliance on collecting realistic training videos. VIVIDCAM incorporates multiple disentanglement strategies that isolate camera motion learning from synthetic appearance artifacts, ensuring more robust motion representation and mitigating domain shift. We show that our design synthesizes a wide range of precisely controlled camera motions using surprisingly simple synthetic data. Notably, this synthetic data often consists of basic geometries within a low-poly 3D scene and can be efficiently rendered by engines like Unity. Our video results can be found in https://wuqiuche.github.io/VividCamDemoPage/.

*Part of this work was completed during Qiucheng's internship at Adobe under the supervision of Handong Zhao. [1]UC, Santa Barbara [2]Adobe Research [3]MIT-IBM Watson AI Lab. Correspondence to: Qiucheng Wu <qiucheng@ucsb.edu>, Handong Zhao <hazhao@adobe.com>.

# 1. Introduction

In creative video generation, camera motion is pivotal for conveying intent, enhancing expressivity, and adding artistic value. As a result, recent work in video generation has focused heavily on equipping text-to-video models with camera control. Camera control in video generation is commonly approached via two paradigms: *text-based control,* where motion is described directly in the input prompt (Liu et al., 2024; Artlist, 2025; Google, 2025), and *trajectory-based control,* where explicit 3D motion trajectories are provided as additional conditioning (He et al., 2024a; Bahmani et al., 2025; Wang et al., 2025). With sufficient training data labeled with camera motion, both paradigms can effectively reproduce similar motion patterns in generated videos.

However, truly creative video generation demands more than just replicating conventional camera techniques like panning or dollying. It requires inventing stylized, intricate motions (as exemplified by the dramatic impact of the *Dolly Zoom* in Hitchcock's *Vertigo*), or crafting scene-specific movements tailored to expressive content (e.g., tracking an unconventional car race). In such cases, collecting enough training data that embodies these avant-garde or bespoke camera motions is infeasible.

Unfortunately, without enough training data support, neither the text-based nor trajectory-based camera control can generalize well to unseen camera motions. For example, Figure 2 illustrates results from state-of-the-art baseline method (Bahmani et al., 2025): while it can reproduce conventional motions such as a forward push ($1^{st}$ row), it fails on more complex, expressive ones ($2^{nd}$ row). In this case, the intended motion was *the camera pans left and right, seeking a sunflower in a glass vase, then locking focus and pushing in upon finding it* (full video: https://wuqiuche. github.io/VividCamDemoPage/). As shown, the generated videos fail to follow the delicate control of the camera panning left/right process, resulting in videos with large perturbations and losing focus.

In short, there is a fundamental paradox between the data-intensive nature of training generative AI models and the creative nature of camera control. So our question is: Can we enable learning out-of-distribution camera motions without real training data?

In this paper, we explore an alternative solution: instead of collecting real videos with uncommon camera motions, we *generate synthetic videos* with the intended motions as training data for generative models. However, while synthetic videos can cover arbitrary motions, they often exhibit virtual styles that diverge significantly from realistic ones. Directly training models on such videos would seriously degrade generation quality. Of course, such drawbacks could potentially be alleviated by generating super-high-quality,

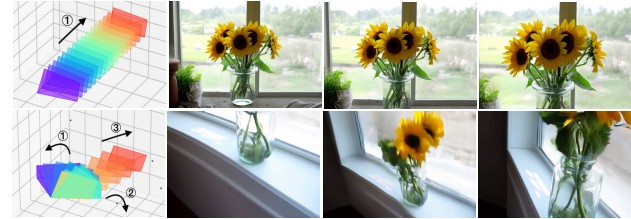

*Figure 2.* State-of-the-art method (Bahmani et al., 2025) fails to generate unconventional camera motions. More examples are in Appendix D.

real-like synthetic videos (Shuai et al., 2025), but this would incur tremendous manual efforts and professional expertise to prepare these videos.

Essentially, the problem boils down to a *disentanglement problem*, *i.e.,* separating appearance and camera motion information in synthetic videos and guiding models to learn only the latter. To this end, we present VIVIDCAM, which uses synthetic **Vir**tual **Vid**eos to fine-tune models for producing correct **Cam**era motions. VIVIDCAM focuses on disentanglement mechanisms. First, inspired by AnimateDiff (Guo et al., 2024), we adopt a dual-adaptation training scheme: we first learn the appearance of synthetic videos through a LoRA and then learn the camera motion; at inference, the appearance LoRA is discarded so the outputs no longer carry undesirable virtual styles. However, this technique has only been used to bridge minor appearance gaps, and is insufficient to resolve the drastic appearance differences between realistic and virtual videos. To further mitigate virtual appearance, VIVIDCAM employs a training recipe with two complementary components: (i) *Data*: we synthesize two sets of videos for training, one without camera motions (to train the appearance LoRA) and one with motions (to train the motion module); and (ii) *Training signals*: we introduce an temporal derivative consistency loss for the motion module, providing appearance-invariant supervision that stabilizes motion learning and strengthens disentanglement. Finally, we use style-aligned text prompts to *anchor* virtual appearance, enabling the model to better distinguish virtual from realistic styles.

We find that even when the generated videos are of very low visual quality, comprising only basic geometries rendered in low-poly 3D scenes (top row in Fig. 1), VIVIDCAM can still effectively disentangle the synthetic appearance from motion and generate realistic videos with the camera motion learned from the virtual videos (bottom four rows in Fig. 1). Our experiments demonstrate that models trained with VIVIDCAM can handle a wide variety of complex, compound camera motions while maintaining realistic visual quality comparable to models trained on real footage.

In summary, our main contributions are as follows:

1. We introduce VIVIDCAM, a novel framework for generating realistic videos with diverse camera motions by leveraging synthetic data efficiently rendered from engines like Unity.

2. Despite the significant domain gap, VIVIDCAM effectively mitigates artifacts present in low-quality synthetic videos and focuses on learning their complex camera motions.

3. We demonstrate that our framework can synthesize a wide range of camera motions with precise, consistent control and high visual quality.

## 2. Related Work

### 2.1. Video Generative Models

Early explorations in video generation tasks focused on GANs (Saito et al., 2017; Tulyakov et al., 2018). Recently, leveraging advancements in diffusion models (Ho et al., 2020; Song et al., 2021; Peebles & Xie, 2023; Ma et al., 2024a), video generative models have rapidly evolved. Early diffusion-based video generative models originated from adapting existing image generative models by incorporating additional motion modules (Guo et al., 2024; Blattmann et al., 2023b; Gu et al., 2023; Singer et al., 2022; Wu et al., 2023). More recently, many end-to-end video generation models have achieved superior results in terms of video quality, resolution, and duration (Blattmann et al., 2023a; Hong et al., 2022; Yang et al., 2025; Ma et al., 2024b). For conditional video generation, various works have explored using text or images to guide the content in generated videos. Rapid progress in video generation has led to several groundbreaking works (OpenAI, 2024; Wang et al., 2024a; Kuaishou, 2025; Google, 2025).

### 2.2. Camera Control in Video Generation

Recently, a line of research has focused on enhancing the controllability of video generative models (Sun et al., 2025; Zhou et al., 2025; Peng et al., 2024). An important aspect is managing camera motion (Zhang et al., 2025a; Xu et al., 2024a; Hou & Chen, 2024; Ling et al., 2024). Early works learned simple, fixed movements (e.g., zooming, panning) from reference videos (Guo et al., 2024; Blattmann et al., 2023a), while later methods conditioned generation on input trajectories (Xu et al., 2024b; Yang et al., 2024), representing cameras through camera matrices (Wang et al., 2024c) or Plücker embedding (He et al., 2024a; Bahmani et al., 2025). These approaches, however, depend on large annotated datasets, which are scarce and offer only limited motion diversity. Recent works (He et al., 2025; Yu et al., 2025; Wang et al., 2025) have devoted significant effort to constructing and curating large-scale realistic video datasets with camera trajectory annotations.

### 2.3. Improving Video Generation With Synthetic Data

Training on realistic video datasets often faces limitations, such as the absence of camera motion annotations and the limited diversity of motion patterns (He et al., 2024a; Wang et al., 2024c). As such, several studies turned to synthetic datasets, many of which focus on multi-view generation (Bai et al., 2025b) and human animation (Black et al., 2023; Wang et al., 2024b; Yang et al., 2023). For camera motion editing, recent work has synthesized new training videos by modifying camera trajectories in existing sequences (Bai et al., 2025a), though these efforts are generally restricted to simple motions. For camera control in video generation, recent approaches generate training videos with explicit camera trajectories in virtual scenes rendered by 3D engines (Fu et al., 2025; Shuai et al., 2025). However, these methods typically demand substantial manual effort to generate complex and diverse scenes and objects. In contrast, our work leverages simple, low-poly 3D environments, enabling diverse camera motions without reliance on large-scale annotation or labor-intensive scene design.

## 3. Preliminaries

We first provide a brief overview of text-to-video diffusion models, which serve as the base model of our work. To train a diffusion model, we first generate a set of corrupted videos, denoted as $X_{1:T}$, by adding progressively increasing Gaussian noises $\epsilon_{1:T}$, to the clean video, $X_0$. The diffusion model then learns to predict the additive noise and denoise noisy videos into cleaner videos. We focus on two types of diffusion models with slightly different training objectives.

**Text-Based Control Only.** Standard text-to-video diffusion models condition the denoising process only on a text input, denoted as $c$. The training loss can be written as

$$\mathcal{L}(\boldsymbol{\theta}_d) = \mathbb{E}_{X_0, t, \epsilon_t}[\epsilon_t - \hat{\boldsymbol{\epsilon}}_{\boldsymbol{\theta}_d}(X_t, c, t)], \tag{1}$$

where $\boldsymbol{\theta}_d$ denotes the parameters of the noise predictor. During inference, clean videos are progressively denoised from pure-noise videos using the trained noise predictor conditional on text $c$. We denote the generation process as $\boldsymbol{g}_{\boldsymbol{\theta}_d}(c)$.

**Trajectory-Based Control.** Recent works control camera poses using trajectories represented as Plücker embeddings $p$, which are encoded via $E_{\boldsymbol{\theta}_e}(p)$ with parameters $\boldsymbol{\theta}_e$. The training loss becomes

$$\mathcal{L}(\boldsymbol{\theta}_d, \boldsymbol{\theta}_e) = \mathbb{E}_{X_0, t, \epsilon_t}[\epsilon_t - \hat{\boldsymbol{\epsilon}}_{\boldsymbol{\theta}_d}(X_t, c, E_{\boldsymbol{\theta}_e}(p), t)]. \tag{2}$$

The inference generation process becomes $\boldsymbol{g}_{\boldsymbol{\theta}_d}(c, E_{\boldsymbol{\theta}_e}(p))$.

## 4. VIVIDCAM

In this section, we first formulate the research problem of synthesizing unconventional camera motions. Then, we in-

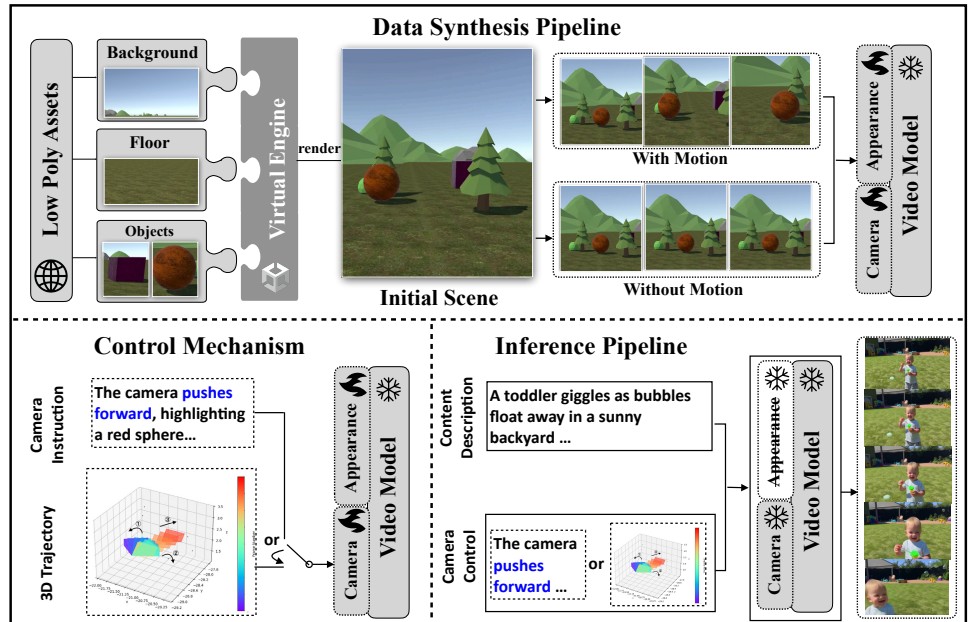

*Figure 3.* Overview of VIVIDCAM. Above: Data Synthesis Pipeline. We first render the initial scene using publicly available assets. Then, we render videos with and without camera motion. These videos are used to train the camera and appearance modules, generating videos with desired camera motions. Bottom-left: Control Mechanism. Our camera control can be achieved by either camera text instruction or a 3D trajectory as input. Bottom-right: Inference Pipeline. During inference, the appearance LoRA is dropped.

troduce VIVIDCAM, a framework to generate them leveraging completely virtual synthetic videos. The overall pipeline is shown in Fig. 3.

### 4.1. Problem Formulation

We aim to fine-tune diffusion models to follow certain camera motions, which can be unconventional so the pre-trained models do not generalize well on them.

Specifically, we consider two camera control paradigms. For **text-based control**, the camera motion instructions are specified as additional input prompts, denoted as $c_m$, such as *'The camera pans around, searching for a bird.'* We fine-tune a *text-only diffusion model*, parameterized as $\theta_d$, to integrate the additional camera instructions, so the generation process becomes $g_{\theta_d + \Delta\theta_{cd}}(c_m \oplus c)$, where $\oplus$ denotes text concatenation, and $\Delta\theta_{cd}$ denotes the fine-tuning weight difference.

For **trajectory-based control**, the camera motions are in the form of out-of-distribution camera trajectories $p$. We fine-tune only the *trajectory encoder* of a text-to-video diffusion model, since the encoder already provides basic camera control abilities (Bahmani et al., 2025). The adapted generation process becomes $g_{\theta_d}(c, E_{\theta_e + \Delta\theta_{ce}}(p))$, where $\Delta\theta_{ce}$ are fine-tuning weight difference (only the trajectory encoder weights are updated; the diffusion model is kept frozen).

Since the camera motions are unconventional, realistic video

datasets often lack sufficient examples for training. We propose to leverage synthetic videos rendered by a physics engine, which allows for arbitrary and diverse camera trajectories. As such, we consider two research questions:

• What types of synthetic videos are most effective for enhancing camera motion learning?

• How to ensure the model learns camera motion independently of the virtual video's appearance?

We first detail the process for rendering training videos in Sec. 4.2. Then, we present our pipeline for disentangling camera motion from artificial appearances in Sec. 4.3.

### 4.2. Render Training Videos

Our first step is to prepare synthetic videos for camera motion training. Leveraging the rendering engine, we are able to generate arbitrary camera motions within a virtual scene. Note that while it is possible to include a variety of synthetic objects in the scene, we focus on constructing the scene with **minimal** numbers and categories of objects to reduce human efforts. Below, we describe the key details of the rendering process. More details can be found in Appendix A.

**The rendered scene.** All synthetic videos are rendered in a low-poly 3D scene using Unity. The scene consists of a *background*, *floor*, and *objects*. Examples of these elements are shown in Fig. 3. Notably, these elements are created using basic geometries. When preparing to synthesize videos,

| Category | | Examples | Videos/Filmmaking Applications |
|---|---|---|---|
| **Simple** | | Push in, Tilt up | ❒ Emphasize an object |
| **Composed** | | Push in → Truck left | ❒ Emphasize, then reveal surroundings |
| **Complex** | Expressive | Seek object (pan/tilt search) | ❒ Simulate searching for a target |
| | | Switch focus between objects | ❒ Highlight relational dynamics |
| | | Orbit shot | ❒ Showcase all sides of objects |
| | | Handheld shake | ❒ Simulate instability or realism |
| | Stylized | Dolly zoom | ❒ Create dramatic perspective shift |
| | | Explosive shake | ❒ Convey impact or chaos |
| | | Camera rotation (90/180°) | ❒ Disorient viewer, mark transition |

*Table 1.* Summary of camera motion categories. We classify them into three categories based on their complexity.

we first randomly sample a background, floor texture, and arbitrarily determine the positions of the objects. Fig. 3 also illustrates an example of the initial settings of the scene.

**Videos with camera motions.** Next, we define the camera motion for the videos. Unity allows users to specify camera movement through code, enabling the simulation of arbitrary camera motions in the scene with just a few lines of code. We consider diverse simple and complex motions listed in Table 1. We denote these synthetic videos with camera motions as $\mathcal{X}_c$ ($c$ stands for 'camera').

**Videos without camera motions.** VIVIDCAM also requires a set of videos without camera motions to aid the disentanglement between appearance and camera motion. Therefore, as shown in Fig. 3, we synthesize another set of training videos, denoted as $\mathcal{X}_a$ ($a$ stands for 'appearance'), which consist of identical appearance styles using a static camera.

### 4.3. Learning Camera Motion from Synthetic Videos

Next, we introduce the pipeline to learn camera motions without introducing synthetic side effects by leveraging the rendered data $\mathcal{X}_c$ and $\mathcal{X}_a$. Prior work suggests that a LoRA module trained specifically to capture the appearance of videos can help mitigate the domain gap in realistic video training datasets (Guo et al., 2024). Motivated by this insight, we investigate whether similar techniques can be applied to absorb synthetic artifacts in *fully virtual-style videos*. Our training involves two steps.

● *Step 1: Appearance Adaptation.* We first train a LoRA, called the *appearance LoRA*, to model the visual characteristics of synthetic scenes without entangling motion, using the static videos $\mathcal{X}_a$ that contain no camera movement. The training objective is thus formulated as

$$\mathcal{L}_{\text{appear}}^{(\text{txt})}(\Delta\boldsymbol{\theta}_a) = \mathbb{E}_{\boldsymbol{X}_0,\,t,\,\boldsymbol{\epsilon}_t}\left[\boldsymbol{\epsilon}_t - \hat{\boldsymbol{\epsilon}}_{\boldsymbol{\theta}_d + \Delta\boldsymbol{\theta}_a}(\boldsymbol{X}_t, \boldsymbol{c}, t)\right],$$
$$\mathcal{L}_{\text{appear}}^{(\text{traj})}(\Delta\boldsymbol{\theta}_a) = \mathbb{E}_{\boldsymbol{X}_0,\,t,\,\boldsymbol{\epsilon}_t}\left[\boldsymbol{\epsilon}_t - \hat{\boldsymbol{\epsilon}}_{\boldsymbol{\theta}_d + \Delta\boldsymbol{\theta}_a}(\boldsymbol{X}_t, \boldsymbol{c}, E_{\boldsymbol{\theta}_e}(\boldsymbol{p}_0), t)\right].$$
$$(3)$$

where $\boldsymbol{X}_0$ are the videos sampled from $\mathcal{X}_a$ dataset, $\Delta\boldsymbol{\theta}_a$ denote the appearance LoRA parameters, $\boldsymbol{c}$ only contains scene descriptions (and no camera motion descriptions), and

$\boldsymbol{p}_0$ denotes a static camera trajectory.

● *Step 2: Camera Control Learning.* We then further fine-tune the models on the new camera motion, in the presence of the trained appearance LoRA (which is kept frozen), using the dataset $\mathcal{X}_c$. The training objective consists of two components. The first is the standard diffusion loss:

$$\mathcal{L}_{\text{diff}}^{(\text{txt})}(\Delta\boldsymbol{\theta}_{cd}) = \mathbb{E}_{\boldsymbol{X}_0,t,\boldsymbol{\epsilon}_t}\left[\boldsymbol{\epsilon}_t - \hat{\boldsymbol{\epsilon}}_{\boldsymbol{\theta}_d + \Delta\boldsymbol{\theta}_a + \Delta\boldsymbol{\theta}_{cd}}(\boldsymbol{X}_t, \boldsymbol{c}_m \oplus \boldsymbol{c}, t)\right],$$
$$\mathcal{L}_{\text{diff}}^{(\text{traj})}(\Delta\boldsymbol{\theta}_{ce}) = \mathbb{E}_{\boldsymbol{X}_0,t,\boldsymbol{\epsilon}_t}\left[\boldsymbol{\epsilon}_t - \hat{\boldsymbol{\epsilon}}_{\boldsymbol{\theta}_d + \Delta\boldsymbol{\theta}_a}(\boldsymbol{X}_t, \boldsymbol{c}, E_{\boldsymbol{\theta}_e + \Delta\boldsymbol{\theta}_{ce}}(\boldsymbol{p}), t)\right].$$
$$(4)$$

where $\boldsymbol{X}_0$ are the videos sampled from $\mathcal{X}_c$ dataset, $\boldsymbol{c}_m$ is the camera motion description. $\Delta\boldsymbol{\theta}_{cd}$ is a LoRA module only used for text-based model. For trajectory-based control, instead of LoRA fine-tuning, we perform full fine-tuning on the camera encoder, denoted by $\Delta\boldsymbol{\theta}_{ce}$.

In addition, we introduce an *temporal derivative consistency loss* to encourage camera motion learning by aligning frame-to-frame differences between predictions and ground truth:

$$\mathcal{L}_{\text{flow}} = \frac{1}{K-1}\sum_{k=1}^{K-1}\left\|\left(\hat{\boldsymbol{X}}_0^{(k+1)} - \hat{\boldsymbol{X}}_0^{(k)}\right) - \left(\boldsymbol{X}_0^{(k+1)} - \boldsymbol{X}_0^{(k)}\right)\right\|_1,$$
$$(5)$$

where $\hat{\boldsymbol{X}}_0$ is the predicted video reconstructed from the noisy input $\boldsymbol{X}_t$; $(k)$ denotes the frame index. The final training loss in step 2 is therefore $\mathcal{L} = \mathcal{L}_{\text{diff}} + \lambda\mathcal{L}_{\text{flow}}$, where $\lambda > 0$ balances two losses.

It is important to note that the camera control adaptation is performed on different model components for text- and trajectory-based methods. For text-based control, the camera LoRA is performed on the diffusion model weights, superimposed on top of the appearance LoRA. For trajectory-based control, the adaptation is performed on the trajectory encoder, which is separated from the appearance LoRA on the diffusion model, because the former is responsible for processing the trajectory information.

The fundamental idea behind dual adaptation is that since the appearance LoRA already learns the virtual appearance information, the camera control learning no longer needs to learn the same information, and can focus on what the appearance LoRA does not learn – camera motion.

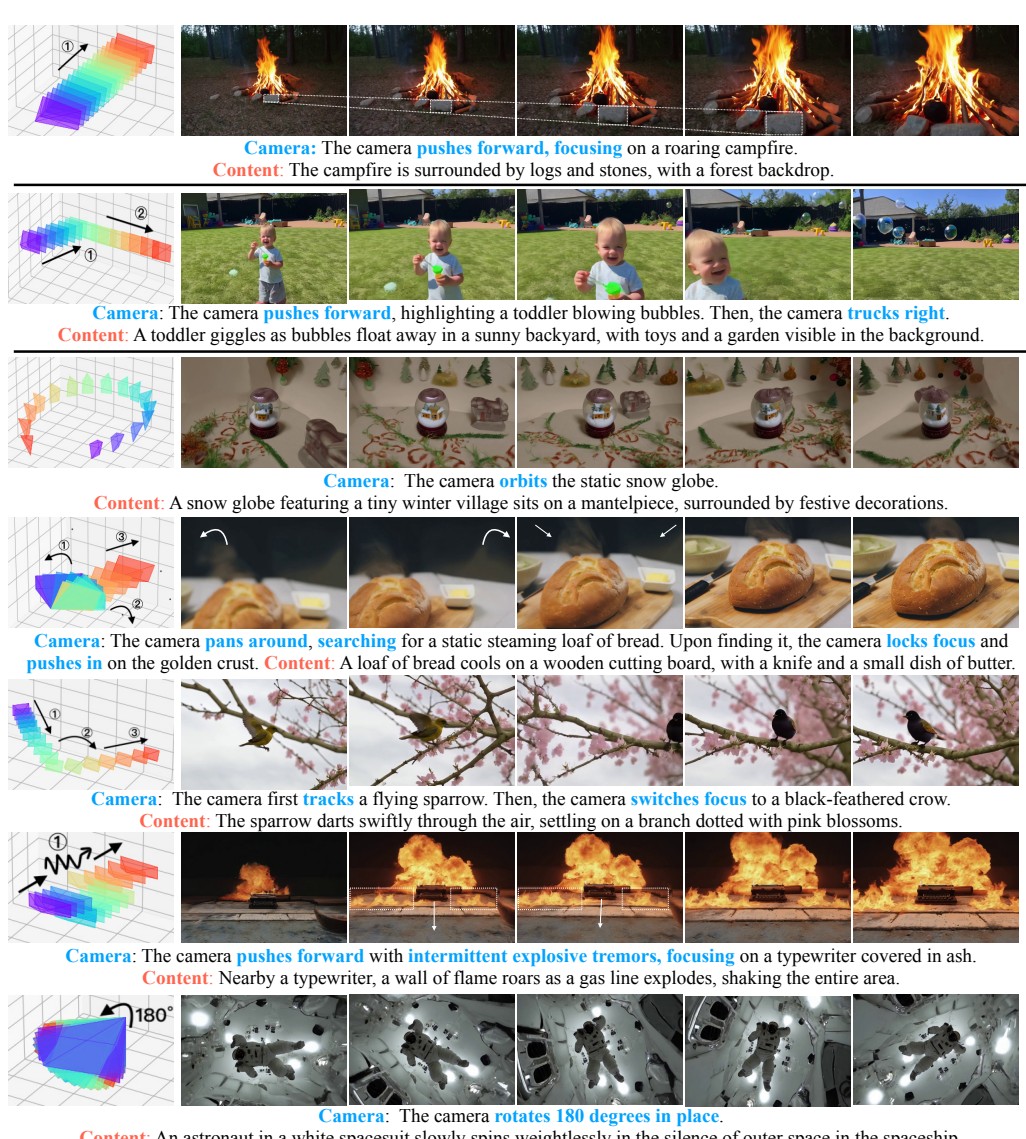

*Figure 4.* Qualitative results. From top to down: ❶ Push forward; ❷ Push forward, then truck right; ❸ Orbit shot; ❹ Pan around, then focus on one object; ❺ Switch focus between objects; ❻ Camera shaking; ❼ Camera rotating. Note that some complex camera motions are difficult to demonstrate through images; please refer to the videos on our webpage `https://wuqiuche.github.io/VividCamDemoPage/` for better visual results. More results can be found in Appendix D.

• *Inference.* After the training, we only deploy camera modules $\Delta\theta_{cd}$ and $\Delta\theta_{ce}$ during inference, while the appearance LoRA $\Delta\theta_a$ is discarded. This would largely remove the undesirable synthetic appearance acquired during training. Specifically, given an input text prompt $c$ with camera instruction $c_m$ or camera pose $p$, the video can be synthesized using $g_{\theta_d + \Delta\theta_{cd}}(c_m \oplus c)$ for text-based model or $g_{\theta_d}(c, E_{\theta_e + \Delta\theta_{ce}}(p))$ for trajectory-based model, respectively. A more detailed description and examples of the training and inference prompts can be found in Appendix B.

**Style-aligned Prompt.** While the appearance LoRA helps address domain gaps, we observe that relying on it alone still introduces synthetic artifacts (see Sec. 5.4). To further

disentangle the appearance and camera motion learning, during both training stages, which are trained on virtual videos, we append a virtual indicator to the input text prompt, $c$, in the form of *'In this low-poly* <VIRTUAL> *scene.'* This would help the model differentiate the virtual style. During inference, this virtual indicator is dropped, which further removes the virtual appearance quality.

## 5. Experiments

In this section, we evaluate VIVIDCAM under various camera motions and compare it with state-of-the-art methods. We focus on the following questions: ❶ What types of camera motions can VIVIDCAM generate? ❷ Can VIVIDCAM

| | Simple Motion | | | Composed Motion | | | Complex Motion | | |
|---|---|---|---|---|---|---|---|---|---|
| | TransErr ↓ | RotErr ↓ | FVD ↓ | TransErr ↓ | RotErr ↓ | FVD ↓ | TransErr ↓ | RotErr ↓ | FVD ↓ |
| CAMERACTRL (He et al., 2024a) | 0.3578 | 0.1358 | 2577.35 | 0.3835 | 0.1989 | 2000.37 | 0.5753 | 0.7042 | 1503.76 |
| COGVIDEOX (Yang et al., 2025) | 0.3113 | **0.0297** | 1781.90 | 0.5107 | **0.1338** | 2168.99 | 0.4327 | 0.5067 | 1488.36 |
| AC3D (Bahmani et al., 2025) | 0.2639 | 0.0973 | 1996.32 | 0.4389 | 0.1958 | 2241.88 | 0.4271 | 0.5864 | 1719.82 |
| VIVIDCAM -COG | **0.1704** | 0.0407 | 1808.30 | 0.2208 | 0.1593 | 2162.60 | 0.4011 | 0.5013 | 1866.40 |
| VIVIDCAM -AC3D | 0.2502 | 0.1162 | 2007.15 | **0.1908** | 0.1968 | 2280.71 | **0.3376** | **0.3619** | 1721.45 |

*Table 2.* Camera pose precision measurements. The best TransErr and RotErr values are in **bold**, and the second-best are underlined.

| | Simple Motion | | Composed Motion | | Complex Motion | |
|---|---|---|---|---|---|---|
| | Action Correctness ↑ | Realism ↑ | Action Correctness ↑ | Realism ↑ | Action Correctness ↑ | Realism ↑ |
| CAMERACTRL (He et al., 2024a) | 0.79 | 0.68 | 0.90 | 0.74 | 0.62 | 0.60 |
| COGVIDEOX (Yang et al., 2025) | 0.80 | 0.74 | 0.53 | 0.65 | 0.61 | **0.68** |
| AC3D (Bahmani et al., 2025) | 0.74 | 0.72 | 0.75 | 0.66 | 0.68 | 0.65 |
| VIVIDCAM -COG | **0.86** | 0.77 | **0.93** | 0.74 | 0.77 | **0.68** |
| VIVIDCAM -AC3D | 0.74 | **0.78** | 0.90 | **0.78** | **0.81** | 0.65 |

*Table 3.* Human study results. The best scores are shown in **bold**, and the second-best are underlined.

provide precise camera control? ❸ Despite being trained exclusively on low-poly synthetic videos, can VIVIDCAM synthesize high-quality realistic videos?

## 5.1. Experiment Settings

**Implementation.** For *text-based control*, we use `CogVideoX-5B` (Yang et al., 2025) as the base model. Notably, different camera motions are trained using a single camera LoRA module. For *trajectory-based control*, we fine-tune `AC3D` built on `CogVideoX-5B`. Similarly, a single trajectory encoder is trained to handle multiple motion types. More details are in Appendix C.

**Camera Motions.** As shown in Table 1, we systematically consider three broad categories of camera motions: *simple*, *composed*, and *complex* movements. *Simple* camera movements refer to basic motions in six different directions. *Composed* movements combine two simple motions. Specifically, when motion 1 is combined with motion 2, the first half of the video follows motion 1, and the second half motion 2. We consider combinations of {push in, pull out} × {truck left, truck right}, resulting in four combinations. *Complex* camera motions include more unconventional practices, divided into *expressive* motions that convey semantic meaning (e.g., seeking objects, handheld shake) and *stylized* motions that create dramatic visual effects (e.g., explosive shake, camera rotation). All of these motions are motivated by realistic applications in everyday videos and filmmaking.

## 5.2. Qualitative Results

We present qualitative results of text-based control in Fig. 4. Please see Appendix D for trajectory-based methods and baselines. We highlight following features:

• **Precise camera motion.** The generated videos exhibit stable camera movements. For example, in row 1, the camera steadily pushes forward at a constant speed, as evidenced by

the predictable changes in the sizes and positions of objects.

• **Ability to handle dynamic objects.** VIVIDCAM can generate precise camera motions for both static and dynamic objects. For example, in rows 1, 2, 5, and 6, we demonstrate that VIVIDCAM can synthesize high-quality moving objects, such as children, birds, and fire effects.

• **Mastery in unconventional camera motions.** VIVIDCAM synthesize unconventional camera motions, many of which semantically depict story-like camera shots. For example, in row 4, VIVIDCAM simulates a common scenario where a person looks around for bread and locks focus once they find it. This capability suggests broad applications beyond simple camera movement. Additionally, we note that some delicate motions, like shaking in row 6, are difficult to convey through static images. We encourage readers to explore our vivid video results on our web page.

## 5.3. Quantitative Evaluations

In this section, we quantitatively compare our framework with existing representative baselines:

• CAMERACTRL (He et al., 2024a) enables camera control via Plücker embedding. In experiments, we provide both prompts and camera poses as inputs.

• COGVIDEOX (Yang et al., 2025) demonstrates text-based camera motion control, likely due to exposure to relative motion data during pre-training. In experiments, we provide prompts that combine both camera motion instructions and content descriptions as inputs.

• AC3D (Bahmani et al., 2025) achieves trajectory-based camera control using ControlNet (Zhang et al., 2023) on DiT (Peebles & Xie, 2023) based models. Similar to CAMERACTRL, we provide the trajectory and prompt as input.

We use the official implementations and checkpoints from their repositories. For fair comparison, we additionally fine-

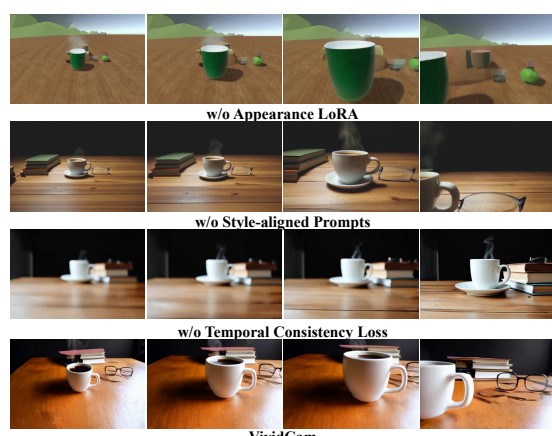

*Figure 5.* Visual examples in ablation study.

|  | Action Correctness | Realism |
|---|---|---|
| Only Dual Adap | 0.83 | 0.61 |
| w/o A-LoRA | 0.75 | 0.60 |
| w/o S-Prompt | 0.93 | 0.65 |
| w/o temp loss | 0.57 | 0.76 |
| ViVidCam | 0.93 | 0.74 |

*Table 4.* Effects of appearance LoRA (A-LoRA) and style-aligned prompts (S-Prompt).

|  | Action Correctness | Realism |
|---|---|---|
| Mixed data | 0.88 | 0.75 |
| Virtual data | 0.91 | 0.72 |

*Table 5.* Effects of realistic data in training.

tune each on our rendered videos under the same data budget. However, we observe a consistent realism collapse due to the synthetic-to-real domain gap in this case, and report these results in Appendix G. We conduct experiments across *simple*, *composed*, and *complex* motions. Each category consists of 100 input prompts. Details can be found in Appendix B.

**Evaluation Metrics.** Following prior work (Cheong et al., 2024; He et al., 2024a), we use *FVD* (Unterthiner et al., 2019) to assess visual quality, and report *TransErr* and *RotErr* to evaluate camera action accuracy. However, in the text-based control setting, the model does not have access to the ground-truth trajectory, making these metrics potentially unfair. To address this, we conduct a *human study* to evaluate both the correctness of camera actions and the realism of the generated videos. We report *Action Correctness* and *Realism* scores from 88 participants (see Appendix F for details). Additional metric results are reported in Appendix I.

**Automated Evaluation Results.** We present the automated metric results in Table 2. Our text-based and trajectory-based methods are denoted as ViVidCam-Cog and ViVidCam-AC3D, respectively. As shown, our method demonstrates strong camera motion precision across diverse motion categories, as indicated by the low TransErr and RotErr values. Additionally, we highlight that our method preserves video quality, as evidenced by the small FVD difference compared to the vanilla CogVideoX and AC3D. Notably, the original CogVideoX achieves good RotErr performance for simple and composed motions. We find this is because such motions involve minimal camera rotation, whereas vanilla CogVideoX typically produces videos with imprecise camera translation (high TransErr) and lacks camera rotation altogether. The following human evaluation further highlights its limitations, and we provide additional discussion of this phenomenon in Appendix D.

**Human study results.** We present human study results in Table 3. We observe that ViVidCam-Cog consistently generates more precise camera motion compared to the base-

lines across all categories of camera motion, and ViVidCam-AC3D shows advantages on complex motions. Additionally, ViVidCam produces high-quality, realistic videos, as evidenced by the comparable realism scores.

### 5.4. Ablation Study

In this section, we conduct ablation studies to examine two key design choices: ❶ **Algorithm side:** How do dual adaptation, temporal consistency loss, and style-aligned prompts affect ViVidCam? ❷ **Data side:** Is the synthetic data generated by VividCam sufficient for training, or does adding realistic videos provide additional benefits?

For algorithm ablation, we conduct human evaluations and report results in Table 4. We highlight following findings:

**Only using dual adaptation cannot successfully disentangle appearance and camera motions.** While prior work (Guo et al., 2024) leverages dual adaptation to reduce the gap between data quality in image and video datasets, we show that this module alone cannot fully bridge the large appearance gap between the synthetic training videos and realistic training videos. As shown in Table 4, using only dual adaptation leads to a clear decline in realism scores.

**The role of each component.** Next, we examine how each component helps mitigate the negative effects of synthetic appearance. Overall, we find that the absence of these components leads to a clear decline in the model's ability to synthesize realistic videos. As shown in Figure 5, without appearance adaptation and style-aligned prompts, the generated appearance closely resembles the synthetic data, resulting in significantly degraded video quality (e.g., synthesized textures in $1^{st}$ row and distorted glasses in $2^{nd}$ row). Meanwhile, without the temporal consistency loss, the model cannot precisely generate the specified camera motions (e.g., $3^{rd}$ row). In contrast, our full model configuration effectively mitigates these visual artifacts, producing clean, realistic videos with the desired camera motions.

**Effectiveness of synthetic data for training ViVidCam.** Third, we examine whether the synthetic videos

generated by ViVidCam provide effective supervision for training, and whether incorporating additional realistic videos further improves performance. We consider the RealEstate10K (Zhou et al., 2018) dataset, which provides annotated camera trajectories, and conduct experiments under the trajectory-based camera control setting. As shown in Table 5, training with only synthetic videos achieves a comparable level of visual realism to training with a mixture of synthetic and realistic videos, with only marginal differences across metrics. Besides, incorporating RealEstate10K does not consistently improve performance and slightly decreases action accuracy. One possible reason is that the camera motions in RealEstate10K are less diverse than those covered by our synthetic data. These results suggest that the synthetic data provides effective supervision for learning camera motion, while preserving the ability to generate realistic videos. Given the substantial human effort required to collect and curate large-scale realistic datasets, ViVidCam offers a practical and cost-efficient training paradigm that can reduce reliance on additional real-world data.

### 5.5. Case Studies and Analysis

In this section, we conduct case studies and additional analyses to better understand the behavior, limitations, and generality of ViVidCam. Qualitative examples are provided on our project webpage.

**Semantic Control with Trajectory Conditions.**  We observe that ViVidCam can combine trajectory control with semantic content descriptions. We provide qualitative examples in our project webpage, "Videos with Semantic Control." In this example, the input trajectory specifies only the camera motion, such as moving left and right, without explicitly defining object positions or the timestamps at which objects should appear. The semantic content is provided through the text prompt, such as a scene containing two birds. As shown in the webpage examples, ViVidCam can arrange the objects according to the specified camera trajectory and generate plausible temporal appearances, even without explicit object placement annotations. This suggests that the model can leverage both the learned video prior and the synthetic trajectory-controlled training data to associate camera motions with reasonable object layouts and movement patterns.

**High Dynamic Scenes.**  We then examine whether ViVidCam can be applied to scenes with stronger object dynamics. Highly dynamic videos remain challenging because the base video generation model does not always reliably synthesize such scenes, often producing artifacts such as ghosting. Since ViVidCam is built on top of the base model, it can inherit these artifacts when the base generation quality degrades. We provide qualitative examples in

our project webpage, "Camera Motions in High Dynamic Scenes." These examples show that the limitation mainly comes from the base model's difficulty in generating highly dynamic content, rather than from the proposed camera-control framework. Therefore, in the main experiments, we primarily focus on dynamic scenes with single-entity interactions, where the base model produces more reliable videos. We provide additional quantitative analysis of highly dynamic scenes in Appendix N.

**Providing Both Text and Trajectory.**  We further study the case where text instructions and 3D camera trajectories are provided together. Our framework supports two camera-control variants: ViVidCam-CogVideoX uses text-based camera control, while ViVidCam-AC3D uses trajectory-based camera control. These two settings are designed separately, since simultaneously using text and trajectory as camera-motion controls may introduce conflicting supervision signals. We show an example of simultaneously providing two signals in our project webpage, "Text-Trajectory Alignment." In this example, we generate videos using a fixed 3D trajectory together with an irrelevant camera-motion instruction, "rotates 180 degrees." The generated results show that the irrelevant text instruction does not noticeably interfere with the trajectory-controlled motion. This indicates that, when a 3D trajectory is provided, ViVidCam-AC3D primarily follows the trajectory condition rather than the camera-motion text.

**Performance on More Base Models.**  In addition to the CogVideoX and AC3D base models, we test ViVidCam on additional base models and domains. Specifically, we consider two new settings: ❶ training on a LoRA-fine-tuned variant of CogVideoX-5B tailored to a different domain, such as cartoon videos; and ❷ training on Wan-AI/Wan2.2-TI2V-5B (Wan et al., 2025), a base model architecturally distinct from CogVideoX. Please see Appendix E for details.

## 6. Conclusion

We propose **ViVidCam**, which uses synthetic **Vi**rtual **Vid**eos to fine-tune video generation models to generate correct **Cam**era motions. Notably, we show that *synthetic videos do not need to be realistic at all*. In fact, ViVidCam shows that video diffusion models can effectively learn camera motion from surprisingly simple synthetic data, often comprising basic geometries rendered in low-poly scenes. Experiments show that models trained with ViVidCam can master various compound and complex camera motions, while maintaining realism comparable to baselines trained on real footage. Ultimately, our work offers an efficient approach to synthesizing realistic videos with precise camera motion control, especially for unconventional motions.

## Acknowledgements

Qiucheng Wu and Shiyu Chang acknowledge the support from National Science Foundation (NSF) Grant IIS-2338252 and Grant IIS-2302730.

## Impact Statement

This paper presents a approach to enable camera motion control in video generation models. Similar to many generative tasks in computer vision, the application of this method would facilitate more convenient usage for end users, allowing them to generate versatile videos with user control. However, if used incorrectly, it could also contribute to the spread of fake videos. Nevertheless, with the help of methods like watermarking, we believe the risks associated with fake videos will ultimately be mitigated.

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

## A. Rendering Training Videos Using Unity

In this work, all training videos are synthesized using Unity. As introduced in Sec. 4.2, for each camera motion synthesis, we first prepare the scene and then render the video with and without camera motion.

**Scene creation.** Each scene consists of a background, a floor texture, and objects. These elements are randomly determined during scene creation. Specifically, we first randomly select the categories of the background and floor texture. The background options include {"sky", "far mountains", "closer mountains", "both mountains"}. The "sky" refers to the default background in Unity. The "far mountains" and "closer mountains" are publicly available background assets that depict mountains at different distances, respectively. The "both mountains" option includes both of the previous mountain backgrounds. The floor texture options include "brick and stone floor", "black sand ground", "green grassland", "brown ground", "yellow grassland", "light green grassland". These textures are also publicly available assets in Unity. For the objects, we randomly place both static and moving objects in the scene. The static objects include "tree", "bush", "grass", while the moving objects include "sphere", "cube", "polygon", "cylinder". Notably, all objects are created using basic geometric shapes and do not require specific human effort for design. Please refer to the example training videos on our website `https://wuqiuche.github.io/VividCamDemoPage/` for a better understanding of their visual appearance.

**Video rendering.** After creating the scene, we render the videos both with and without camera motion. The videos without camera motion are generated by randomly determining the camera's coordinates and pose, then fixing the camera in place while recording the video. For the videos with camera motion, we first define the camera movements using a short script (typically no more than 10 lines of code). Based on this script, the rendered video incorporates the specified camera motions.

## B. Details of Text Prompts

Our training and inference processes rely on different categories of text prompts. In this section, we provide a detailed discussion of the prompts used. Generally, two categories of prompts are employed: (1) scene-only prompts $c$, used for appearance LoRA learning, and (2) composite prompts $(c_m \oplus c)$, which combine camera instructions with scene descriptions for learning camera control in the text-based setting. Additionally, we provide examples of prompts used during inference.

**Scene-Only Prompts for Appearance LoRA Training.** During appearance LoRA training, we constrain the LoRA to learn only the appearance style. Therefore, the training prompt at this stage includes only a description of the rendered scene, specifying objects and environmental details. For example: *"Content: There are small plants and geometries on the light green grassland."* Additionally, as described in Sec. 4.3, we incorporate a style-aligned prompt to help bridge domain gaps during appearance LoRA training. This prompt acts as a virtual indicator of the target style. With this addition, the complete training prompt $c$ becomes, for example: *"Content: In this low-poly 3D `<VIRTUAL>` scene, there are small plants and geometries on the light green grassland."*

**Composite Prompts for Text-Based Camera Control.** For text-based camera control, we freeze the appearance LoRA and train a separate camera LoRA. At this stage, the training prompt includes both camera movement instructions $c_m$ and scene descriptions $c$. The camera component guides the camera LoRA to learn appropriate motion patterns, while the scene description ensures consistent content generation. For example: *"Camera: The camera pushes forward, focusing on a moving sphere. Then the camera trucks left. | Content: In this low-poly 3D `<VIRTUAL>` scene, there is a moving sphere. There are also small plants and geometries on the black sand ground."* It is worth noting that for trajectory-based camera control, we use only the scene description $c$, rather than composite prompts $(c_m \oplus c)$, since the camera condition is provided directly by the trajectory input $p$.

**Prompts at Inference Time.** During inference, we use prompts similar to those employed during camera control training, with the exception that the virtual style indicator is omitted. Below are example prompts for both text-based and trajectory-based control: Text-based: *"Camera: The camera pushes forward, focusing on a static steaming coffee cup. Then the camera trucks right. | Content: A steaming coffee cup rests on a wooden table beside a stack of books and a pair of glasses."* Trajectory-based: *"Content: A steaming coffee cup rests on a wooden table beside a stack of books and a pair of glasses."*

## C. Implementation Details

We use `CogVideoX-5B` (Yang et al., 2025) as the base model. The base model remains frozen throughout all experiments, and we adopt its default hyperparameters (e.g., noise sampling schedule, conditional guidance scale). Each generated video

| | | Value |
|---|---|---|
| **Appearance LoRA** | Learning rate | 1e-4 |
| | Rank | 128 |
| | Scheduler | Cosine with Restarts |
| | Warm up steps | 400 |
| | Optimizer | adamw |
| | $\beta_1$ | 0.9 |
| | $\beta_2$ | 0.95 |
| **Camera LoRA** | Learning rate | 3e-4 |
| | Rank | 512 |
| | Scheduler | Cosine with Restarts |
| | Warm up steps | 400 |
| | Optimizer | adamw |
| | $\beta_1$ | 0.9 |
| | $\beta_2$ | 0.95 |
| **Trajectory Encoder** | Learning rate | 1e-4 |
| | Scheduler | Cosine with Restarts |
| | Warm up steps | 250 |
| | Optimizer | adamw |
| | $\beta_1$ | 0.9 |
| | $\beta_2$ | 0.95 |

*Table 6.* Hyperparameter settings.

is 5 seconds long, consisting of 49 frames at a resolution of $720 \times 480$. For *text-based control*, the learning rate for LoRA optimization is set to 1e-4 for appearance learning and 3e-4 for camera motion learning, with the LoRA rank fixed at 128 and 512 for two LoRA. We use one camera LoRA for different motion types, each using 500 synthetic training videos. For *trajectory-based control*, we fine-tune from the pre-trained AC3D model using a learning rate of 1e-4. We train one encoder for different motion types, using the same set of synthetic training videos as in text-based control.

To help reproduce our results, we report the detailed hyperparameter settings in Table 6.

## D. Qualitative Comparison and Analysis

Section 5.2 presents the qualitative results of the text-based methods (VIVIDCAM-COG). In this section, we first present the qualitative results of the trajectory-based methods (VIVIDCAM-AC3D), followed by a comparison with the baseline methods and corresponding analyses.

**Qualitative results of trajectory-based methods.** We present the qualitative results of VIVIDCAM-AC3D in Figure 6. As shown, similar to the text-based method, our trajectory-based method can generate videos with precise camera control and high visual quality across a range of camera motions, from simple and composed to complex ones.

**Qualitative comparison with baselines.** We present qualitative comparisons in Figure 7 and Figure 8. Our observations indicate that state-of-the-art methods struggle to accurately synthesize unconventional camera motions. For instance, in the upper panel of Figure 7, even when provided with exact trajectories, these methods often simplify the intended motion—resulting in a pan to the right in CAMERACTRL or a turbulent push-in in AC3D. Similarly, in the upper panel of Figure 8, while AC3D attempts to depict a focus shift from a rock to a stone, it fails to effectively illustrate the pull-out from the rock followed by a push-in toward the road. In contrast, our method faithfully captures and reproduces the intended camera motions. Additionally, we note that such unconventional motions are difficult to fully appreciate through static images alone. We encourage readers to refer to the videos on our webpage https://wuqiuche.github.io/VividCamDemoPage/ for better visual results.

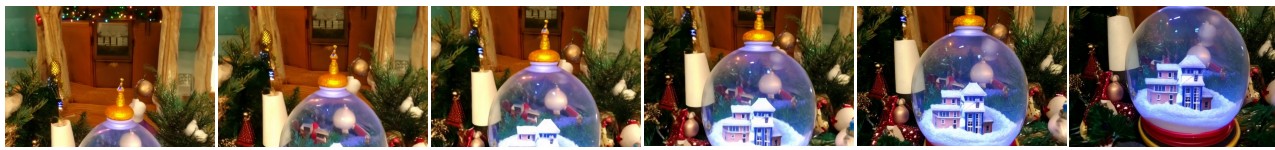

**Camera:** The camera **tilts down, revealing** a snow globe.
**Content**: A snow globe depicting a small town is surrounded by holiday decorations.

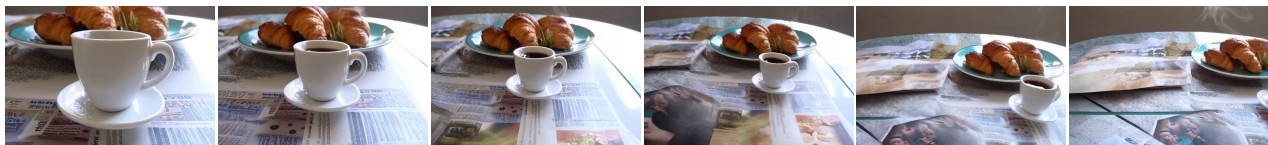

**Camera**: The camera **pulls back**, moving away from a static steaming coffee cup. Then the camera **trucks left**.
**Content**: A steaming coffee cup rests on a glass table beside a plate of croissants and an open newspaper.

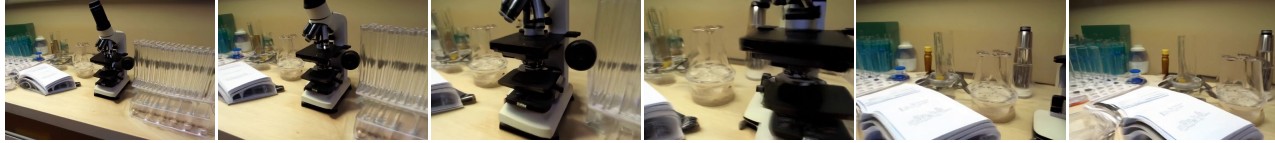

**Camera**: The camera **pushes forward**, focusing on a static microscope. Then, the camera **trucks left**.
**Content**: A laboratory microscope sits on a counter, flanked by glass slides, test tubes, and an open science book.

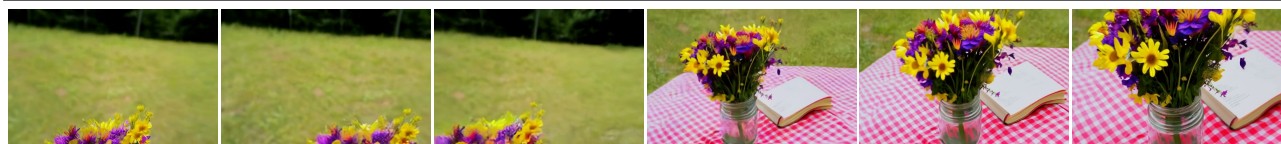

**Camera**: The camera **pans around, searching** for a static bouquet of wildflowers. Upon finding it, the camera **locks focus** and **pushes in** on the petals. **Content**: A bouquet of wildflowers rests in a jar on a picnic table, with a red-checkered cloth beside it.

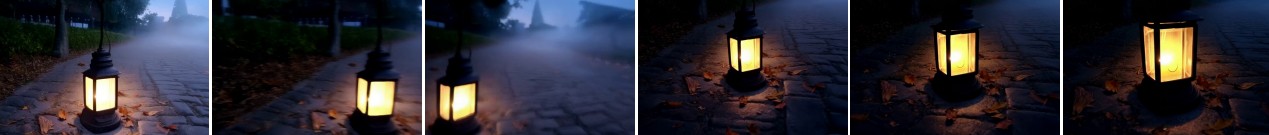

**Camera**: The camera **pans around**, **searching** for a glowing lantern. Upon finding it, the camera **locks focus** and **pushes in** on the warm light inside. **Content**: A lantern glows on a cobblestone path at dusk, surrounded by fallen leaves and a faint mist.

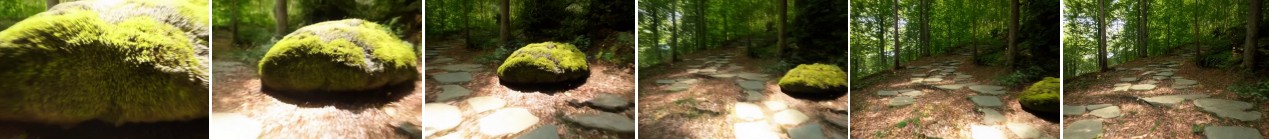

**Camera**: The camera **pulls out** from a moss-covered rock. Then, the camera **zooms in, shifting focus** to a stone pathway.
**Content**: The rock rests under the shade of tall trees, while the pathway winds through the forest, its stones carefully placed.

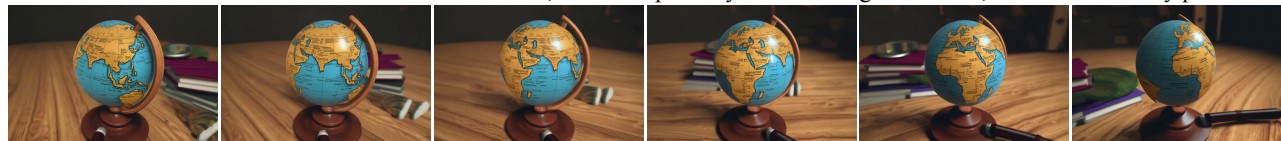

**Camera**: The camera orbits the static globe.
**Content**: A classic globe with detailed continents rests on a wooden stand, surrounded by history books.

*Figure 6.* Qualitative results of diverse camera motions using VIVIDCAM-AC3D. Note that some complex camera motions are difficult to demonstrate through images; please refer to the videos on our webpage https://wuqiuche.github.io/VividCamDemoPage/ for better visual results.

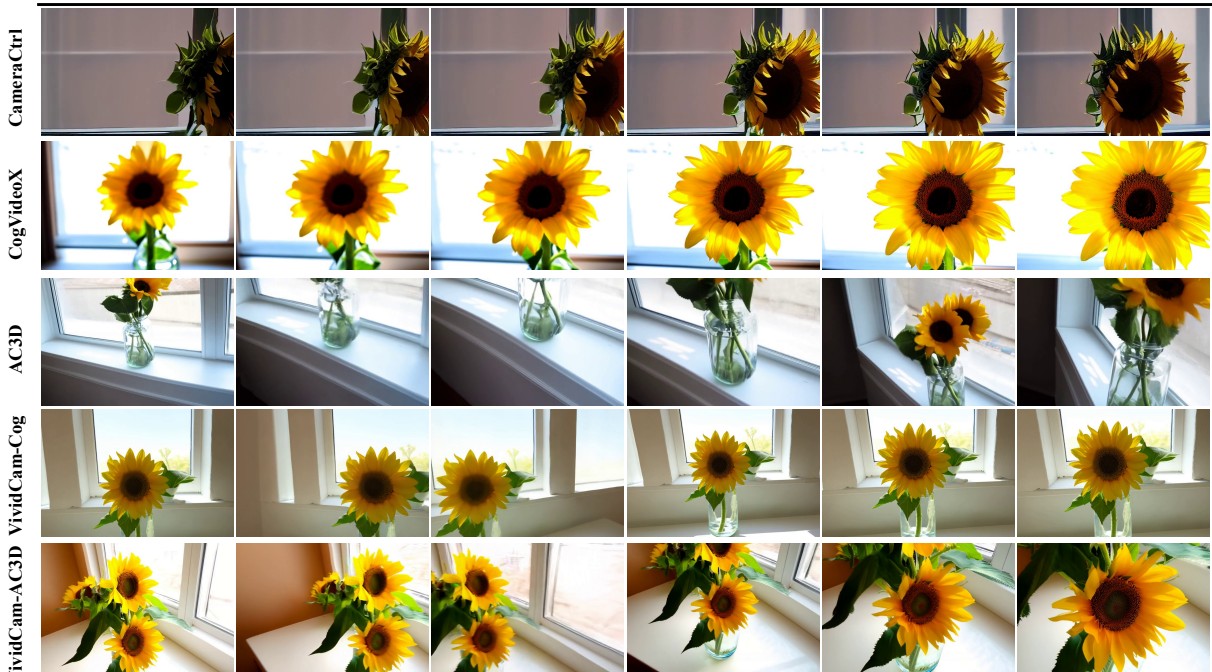

**Camera**: The camera **pans around, searching** for a sunflower in a glass vase. Upon finding it, the camera **locks focus** and **pushes in** on the vibrant yellow petals. **Content**: A sunflower sits in a glass vase on a windowsill, with sunlight streaming in and casting soft shadows on the sill.

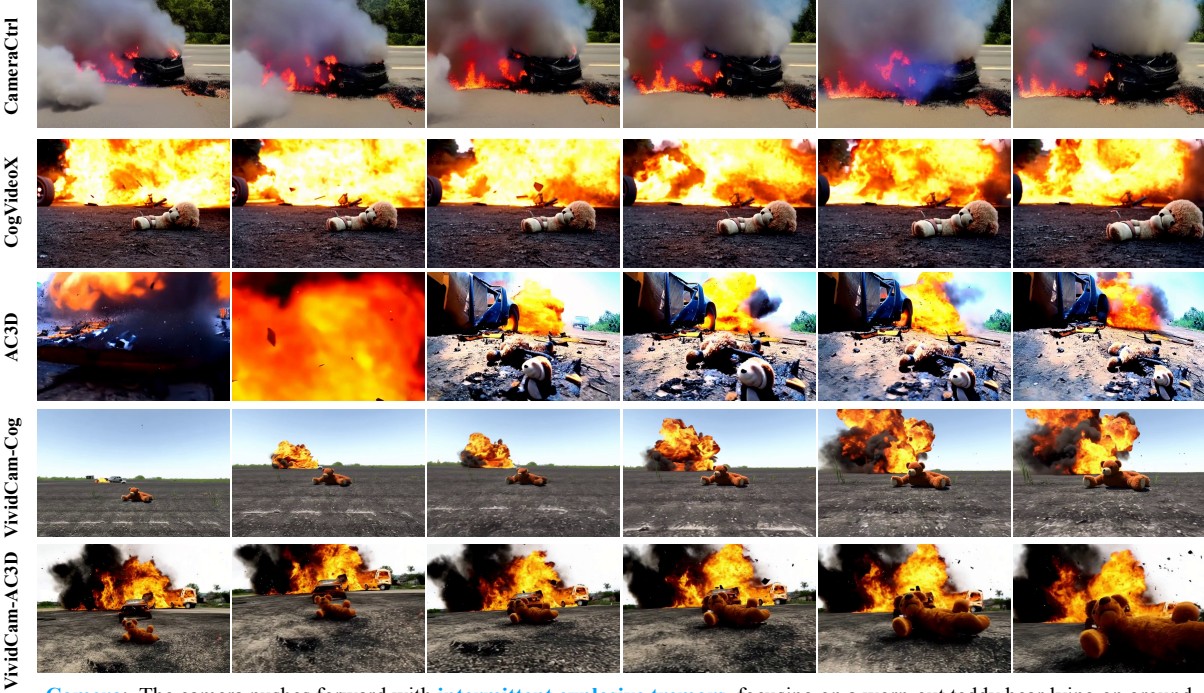

**Camera**: The camera pushes forward with **intermittent explosive tremors**, focusing on a worn-out teddy bear lying on ground. **Content**: A car explodes in a fiery blast, sending debris and dust into the air.

*Figure 7.* Qualitative results comparison. We observe that camera motions such as "panning around to search for an object, then pushing in to focus on the object" are particularly challenging for state-of-the-art models. Even when provided with exact trajectories, these methods often degrade into simpler camera motions—such as a rightward truck in CAMERACTRL or a turbulent push-in in AC3D. In contrast, our method faithfully produces the intended camera motions. Additionally, we note that certain effects, such as explosive camera motions, are difficult to convey through static images. Please refer to the videos on our webpage https://wuqiuche.github.io/VividCamDemoPage/ for better visual results.

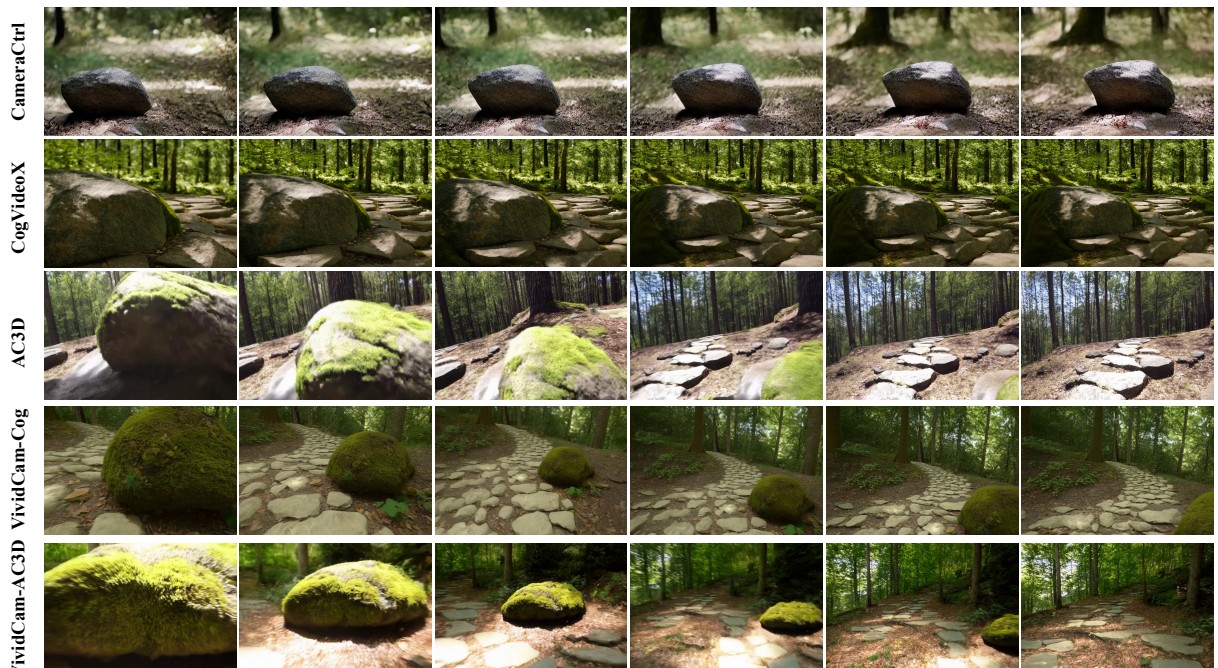

**Camera**: The camera **pulls out** from a moss-covered rock. Then, the camera **zooms in, shifting focus** to a stone pathway.
**Content**: The rock rests under the shade of tall trees, while the pathway winds through the forest, its stones carefully placed.

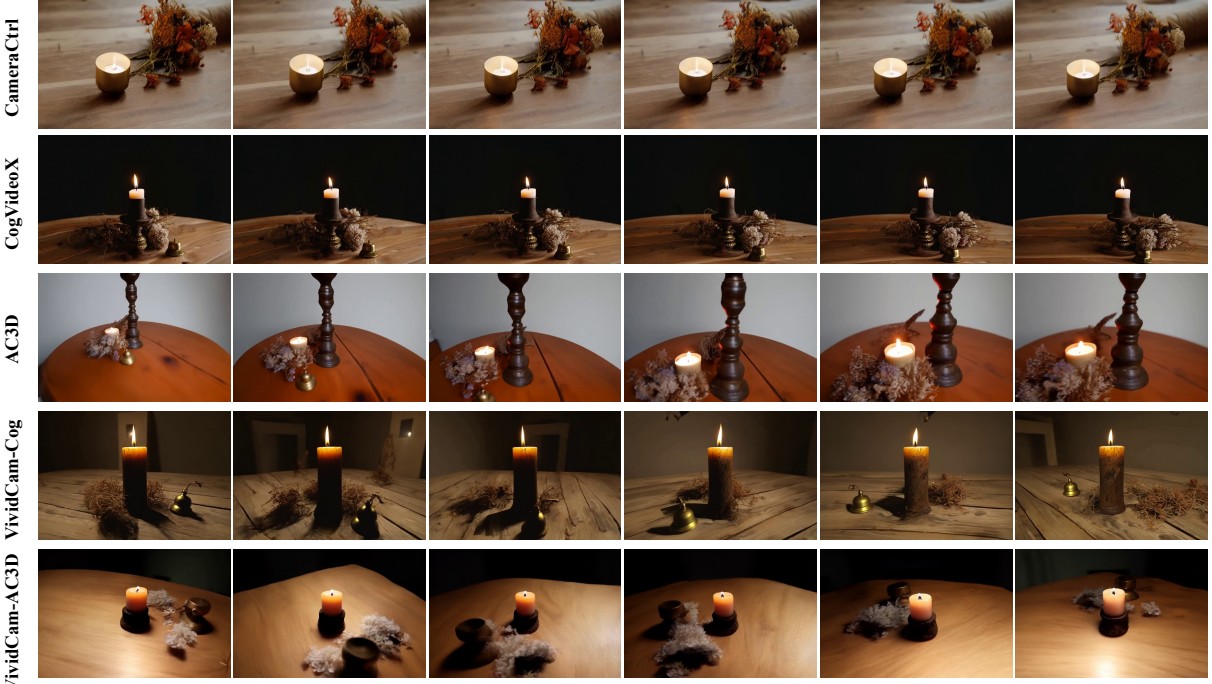

**Camera**: The camera **orbits** the static candle holder.
**Content**: A rustic candle holder with a flickering candle rests on a wooden table, accompanied by dried flowers and a brass bell.

*Figure 8.* Qualitative results comparison. We observe that camera motions such as "pulling out from an object, then zooming in to shift focus to another object" are particularly challenging for state-of-the-art models. Even when provided with exact trajectories, these methods often fail to accurately reproduce the desired camera motions. For example, while AC3D attempts to depict a focus shift from a rock to a stone, it does not successfully demonstrate the pull-out from the rock followed by the push-in toward the road. In contrast, our method faithfully captures and reproduces the intended camera motions. Additionally, we note that such unconventional camera movements are difficult to fully appreciate through static images alone. Please refer to the videos on our webpage https://wuqiuche.github.io/VividCamDemoPage/ for better visual results.

**Instructions:** Please read the instructions carefully. Failure to follow the instructions will lead to the rejection of your results. In this task, you will be asked to evaluate if AI models accurately follow the camera motion instructions in text instruction (e.g. left, right) when generating videos. Specifically, you will first see a text description, which describes the camera motion in the videos we want to generate. The instruction usually consists of two parts movement. Please **pay special attention to the direction**. Next, you will see a short video, which is generated based on the provided text by an AI algorithm. You will then be asked to evaluate if the **camera motion** in the video is consistent with the instruction. Assign 2 points if the camera motion is clear and consistent, 1 point if the camera motion is unclear and consistent with the description in a part of the video, and 0 points if the camera motion is incorrect or no camera motion. Notice that you should **only** focus on the camera motion and ignore any other factors. Additionally, you should ignore the errors where some objects mentioned in the text are missing.

**Example:** We provide an example to help you understand how to evaluate the generated results. The text description is "The camera **pushes forward**, focusing on a moving sphere. Then the camera **pans left.**" To analyze the generated results, we observe that in the video, the camera first pushes forward, then moves leftward from the initial position. The camera motion is clear and consistent, and you should assign 2 points to this video. You should ignore the difference between camera pan and camera truck movements.

**Question:** The text description is **"The camera orbits the static coffee grinder."**.

How is the camera motion in this video consistent with the text description? Assign 2 points if the camera motion is clear and consistent, 1 point if the camera motion is unclear and consistent with the description in a part of the video, and 0 points if the camera motion is incorrect or no camera motion.

○ 0
○ 1
○ 2

Submit

*Figure 9.* The example interface of Amazon Mechinical Turk in our human study.

## E. Generalization to Other Base Models and Domains

Sec. 5 demonstrates the advantages of our method on text-based and trajectory-based control using the base models `CogVideoX` and `AC3D`, both focused on realistic video generation. In this section, we further explore whether VIVIDCAM can generalize to different domains and model architectures. Specifically, we consider two new architectural settings: ❶ training on `Cseti/CogVideoX-LoRA-Wallace_and_Gromit`, a LoRA-fine-tuned variant of CogVideoX-5B tailored to a cartoon domain; and ❷ training on `Wan-AI/Wan2.2-TI2V-5B` (Wan et al., 2025), a base model that is architecturally distinct from CogVideoX.

We present the results in our video link `https://wuqiuche.github.io/VividCamDemoPage/`. In both settings, we observe that VIVIDCAM remains capable of generating unconventional camera motions. These results suggest that VIVIDCAM generalizes well across different base models and domains.

## F. Details of Human Study

Our human study is conducted on Amazon Mechanical Turk. We consider three levels of camera motion: *simple*, *composed*, and *complex*. Please refer to Table 1 for the specific camera motions covered in each category. For each category, we sample 25 prompts and input them into our model and baseline models for evaluation. Each of the 25 prompts is tested twice, resulting in 50 videos per camera motion category and a total of 150 videos. Each question is awarded $0.03. In total, 88 unique workers participate in the study. For each question, we present the tested videos along with the input text prompt and ask participants to answer two types of questions: ❶ (**Action Correctness**) How consistent is the camera motion in the video with the text description? ❷ (**Realism**) How is the visual quality of the video? Participants rate each question on a

| Setting | TransErr ↓ | RotErr ↓ | FVD ↓ |
|---|---|---|---|
| VividCam-Cog | 0.4011 | 0.5013 | 1866.40 |
| VividCam-AC3D | **0.3376** | **0.3619** | 1721.45 |
| CogVideoX | 0.4327 | 0.5067 | **1488.36** |
| CogVideoX + Synthetic FT | 0.3937 | 0.5224 | 2805.18 |
| AC3D | 0.4271 | 0.5864 | 1719.82 |
| AC3D + Synthetic FT | 0.4016 | 0.5358 | 2971.21 |

*Table 7.* Effect of directly fine-tuning baseline models on synthesized videos. While motion errors may slightly decrease, FVD increases significantly, indicating degraded visual quality due to synthetic-to-real domain mismatch. VIVIDCAM avoids this issue through motion–appearance disentanglement.

scale from 0 to 2. To ensure precise evaluation, we provide detailed explanations, a scoring rubric, and examples.

Figure 9 shows an example of the interface that participants will see during the human study.

## G. Fine-tuning Baselines with Synthetic Data

In Section 5, we report results using off-the-shelf checkpoints for all baseline methods, following their official implementations. For a fair comparison, we additionally evaluate a setting where baseline models are directly fine-tuned on the same synthesized videos used to train VIVIDCAM.

As discussed in Section 4, our synthesized videos differ substantially from real-world videos in visual appearance and perceptual statistics (see Figure 3). While such data is effective for learning camera motion patterns, directly training video generation models on this synthetic domain can lead to a mismatch in appearance modeling and temporal realism. As a result, naïve fine-tuning on synthetic videos may degrade visual quality, even if motion-related metrics improve marginally.

To illustrate this effect, we fine-tune each baseline model on exactly the same synthetic video set used by VIVIDCAM, under matched training settings. Table 7 reports the quantitative results. We observe that although motion errors (TransErr and RotErr) occasionally decrease slightly after fine-tuning, the Fréchet Video Distance (FVD) increases substantially, indicating a significant drop in visual realism. This confirms that direct adaptation to synthetic videos introduces severe appearance artifacts and perceptual degradation.

In contrast, VIVIDCAM is specifically designed to disentangle motion learning from appearance modeling, enabling it to leverage synthetic data for camera motion supervision while preserving visual quality at inference time. This experiment further motivates the architectural design choices of VIVIDCAM and explains why we do not fine-tune baseline models on synthetic data in the main comparisons.

## H. CLIP similarity

We present the CLIP similarity results in Table 8. Overall, all methods achieve comparable CLIP similarity scores. Specifically, both VIVIDCAM-COG and VIVIDCAM-AC3D exhibit less than a 0.01 difference in CLIP score compared to their vanilla counterparts, COGVIDEOX and AC3D, respectively. This indicates that our methods maintain the same level of alignment with the desired content described in the text prompts. We also observe

| | Simple | Composed | Complex |
|---|---|---|---|
| CAMERACTRL | 0.3254 | 0.3306 | 0.3006 |
| COGVIDEOX | 0.3272 | 0.3215 | 0.3070 |
| AC3D | 0.3397 | 0.3437 | 0.3153 |
| VIVIDCAM-COG | 0.3327 | 0.3362 | 0.3230 |
| VIVIDCAM-AC3D | 0.3352 | 0.3341 | 0.3134 |

*Table 8.* Results on CLIP similarity.

that COGVIDEOX performs worse in scenarios involving complex camera motions, possibly due to quality degradation when generating motion patterns that are likely underrepresented in the training dataset, as shown in Sec. D.

## I. Additional Metrics for Video Evaluation

In addition to the main evaluation metrics reported in the paper, we further evaluate generated videos from two complementary perspectives: visual quality and dynamic degree. Following VideoScore (He et al., 2024b), the visual quality score measures

the perceptual and aesthetic quality of generated videos, while the dynamic degree score reflects the extent and complexity of temporal changes in the generated video.

Table 9 reports the results. Compared with the corresponding base models, VividCAM achieves higher dynamic degree scores for both CogVideoX and AC3D. This indicates that VividCAM is able to generate videos with more expressive camera motions. At the same time, VividCAM maintains visual quality scores comparable to the corresponding baselines. These results suggest that our method improves camera motion generation without substantially degrading the visual quality of the generated videos.

| Method | Dynamic | Quality |
|---|---|---|
| VIVIDCAM-COG | 2.781 | 2.813 |
| COGVIDEOX (base) | 2.422 | 2.828 |
| VIVIDCAM-AC3D | 3.297 | 2.865 |
| AC3D (base) | 3.187 | 2.841 |
| CAMERACTRL | 2.924 | 3.078 |

*Table 9.* Additional video evaluation using dynamic degree and visual quality scores.

## J. Study on Realistic Rendering Assets

Our main pipeline uses geometry-based synthetic assets to generate large-scale camera-controlled training videos. One natural question is whether using more realistic rendering assets can reduce the train-test domain gap and further improve performance. We view this as a cost-quality trade-off. High-quality assets and rendering pipelines can produce more realistic videos, but scaling such data generation often requires additional cost and curation effort, including asset selection, scene construction, and quality filtering. In contrast, our default setting provides a scalable and low-cost way to generate trajectory-controlled videos.

To further examine this trade-off, we conduct an additional experiment using more realistic assets. Specifically, we collect 10 realistic Unity-compatible assets from online asset stores and use them to render training videos. We compare three settings: our original VIVIDCAM pipeline with geometry assets, directly training on the videos rendered from realistic assets without the VIVIDCAM pipeline, and training with the realistic assets incorporated into the VIVIDCAM pipeline.

As shown in Table 10, directly using videos rendered from realistic assets without the VIVIDCAM pipeline does not lead to better overall performance, and obtains a worse FVD than our original setting. This suggests that simply increasing asset realism is not sufficient for effective camera-control training. When the same realistic

| Setting | TransErr | RotErr | FVD |
|---|---|---|---|
| Original assets | 0.4011 | 0.5013 | 1866.40 |
| Realistic, w/o VIVIDCAM | 0.3822 | 0.5761 | 2125.10 |
| Realistic, w/ VIVIDCAM | 0.4198 | 0.5404 | 1769.27 |

*Table 10.* Results with realistic rendering assets.

assets are used within the VIVIDCAM pipeline, the FVD improves, indicating that higher-quality assets can further improve visual quality when combined with our training design. At the same time, this improvement comes with additional asset cost and curation effort. These results support our choice of geometry-based assets as a practical default setting, while also showing that VIVIDCAM can benefit from higher-quality assets when such resources are available.

## K. Effect of Discarding Appearance LoRA at Inference

During inference, VIVIDCAM discards the appearance LoRA and keeps only the camera-control adaptation. This design follows our appearance-motion disentanglement strategy: the appearance LoRA is introduced to absorb synthetic appearance during training, while the final inference model should preserve the visual prior of the original video generation model. To verify this design, we compare the default inference setting with a variant that keeps the appearance LoRA during inference.

As shown in Table 11, keeping the appearance LoRA during inference substantially degrades video quality for both CogVideoX and AC3D backbones. Specifically, FVD increases for both models. The visual quality score also decreases clearly in both settings. These results suggest that the appearance LoRA indeed captures synthetic-domain appearance information, which is useful for disentangling appearance from camera motion during training but should not be retained at inference time. Therefore, discarding the appearance LoRA is important for preserving the visual quality of the base video generation model.

| Method | FVD | Quality |
|---|---|---|
| VIVIDCAM-COG | 1866.40 | 2.813 |
| with A-LoRA | 2097.61 | 2.280 |
| VIVIDCAM-AC3D | 1721.45 | 2.865 |
| with A-LoRA | 2156.02 | 2.173 |

*Table 11.* Effect of keeping the appearance LoRA during inference.

## L. Effect of Temporal Consistency Weight

We further study the effect of the temporal derivative consistency loss weight $\lambda$ in $\mathcal{L}_{\text{flow}}$. This loss encourages the model to match frame-to-frame changes between the predicted video and the ground-truth video. The weight $\lambda$ controls the balance between camera-motion learning and the original diffusion denoising objective.

As shown in Table 12, a small temporal consistency weight weakens camera-motion learning, leading to worse TransErr and RotErr. Increasing $\lambda$ from 0.03 to 0.3 substantially improves both translation and rotation accuracy. However, using an overly large weight further increases the influence of the temporal consistency objective and leads to worse visual quality, as indicated by the higher FVD. We therefore use $\lambda = 0.3$ as the default setting, which provides a good balance between camera action accuracy and perceptual video quality.

| Setting | TransErr | RotErr | FVD |
|---------|----------|--------|-----|
| $\lambda = 0.03$ | 0.5859 | 0.7156 | 1808.94 |
| $\lambda = 0.3$ | 0.4011 | 0.5013 | 1866.40 |
| $\lambda = 3$ | 0.4266 | 0.4923 | 2124.11 |

*Table 12.* Effect of the temporal consistency weight $\lambda$.

## M. Comparison with a Recent Depth-Conditioned Method

We further compare VIVIDCAM with DualCamCtrl (Zhang et al., 2025b), a recent method for camera-controllable video generation. Unlike VIVIDCAM, DualCamCtrl requires depth images as additional input, which provides extra geometric guidance but also makes the inference pipeline less convenient in practical settings where depth is not readily available. We therefore include this comparison to examine how VIVIDCAM performs against a recent method with a stronger input requirement.

As shown in Table 13, VIVIDCAM achieves lower TransErr and RotErr than DualCamCtrl, indicating more accurate camera-action control. DualCamCtrl obtains a lower FVD, which suggests competitive visual quality under its depth-conditioned setting. Qualitatively, we observe that DualCamCtrl can handle some simple camera motions, such as switching the view between objects, but struggles with more precise instructions such as searching for and focusing on a target object. In contrast, VIVIDCAM produces more consistent camera behaviors across diverse motion types while requiring only text or trajectory conditions.

| Method | TransErr | RotErr | FVD |
|--------|----------|--------|-----|
| VIVIDCAM-COG | 0.4011 | 0.5013 | 1866.40 |
| DUALCAMCTRL | 0.4372 | 0.7134 | 1792.15 |

*Table 13.* Comparison with DualCamCtrl.

## N. Analysis on High Dynamic Scenes

We further quantify whether VIVIDCAM reduces the dynamic generation capability of the base model. Since VIVIDCAM is built on top of CogVideoX, highly dynamic scenes can be challenging when the base model itself fails to generate temporally coherent videos.

To examine this effect, we compare CogVideoX and VIVIDCAM-CogVideoX under both single-object and multi-object dynamic scenes, and report realism scores. As shown in Table 14, VIVIDCAM achieves a comparable realism score to CogVideoX in single-object dynamic scenes and a slightly higher score in multi-object dynamic scenes. These results indicate that our camera-control training does not reduce the base model's

| Method | Single | Multiple |
|--------|--------|----------|
| COGVIDEOX | 2.704 | 2.434 |
| VIVIDCAM-COG | 2.686 | 2.547 |

*Table 14.* Realism scores on dynamic scenes.

ability to generate dynamic content. Together with the qualitative examples in Section 1 of the project webpage, "Camera Motions in High Dynamic Scenes," these results suggest that the remaining failures in highly dynamic scenes mainly arise from the limitations of the base video generation model, rather than from the proposed camera-control framework.

## O. Failure Case Analysis

We further analyze the failure cases of VIVIDCAM. Overall, we observe three main types of failures, corresponding to scene surroundings, dynamic objects, and camera actions. First, when the prompt provides limited descriptions of the surrounding environment, the generated video may contain less realistic backgrounds or incomplete scene details. This suggests that sufficiently descriptive scene prompts remain important for maintaining visual realism. Second, scenes with many dynamic objects can exceed the capacity of the base video generation model. In such cases, the generated objects may become temporally inconsistent or visually incoherent, and this limitation can be inherited by VIVIDCAM since our method builds

on the base model. Third, overly long or complex camera motions are sometimes not fully captured. We conjecture that this is partly due to the limited temporal length of the base model, which generates 49-frame videos and may not have enough temporal capacity to express long-range camera movements. We provide representative visual examples of these failure modes in Section 5 of our project webpage, "Failure Case Examples".

