# OpenReview forum: "VividCam: Learning Unconventional Camera Motions from Virtual Synthetic Videos"
_ICML.cc/2026/Conference — ICML 2026 regular_

### Official Review · Reviewer_hroA · 2026-02-14

**Soundness:** 2
**Presentation:** 3
**Significance:** 3
**Originality:** 3
**Overall Recommendation:** 4
**Confidence:** 2

**Summary:**

The paper proposes VIVIDCAM, which aims to train video generative models to follow unconventional camera motions by using low-poly synthetic videos rendered in engines like Unity. This work is inspired by the two-stage training strategy of AnimateDiff: first adapting to the synthetic style through an appearance LoRA, then training the camera motion module, and finally discarding the appearance LoRA during inference to restore realism. The paper conducts experimental validation under both text-based and trajectory-based camera control paradigms.

**Compliance With Llm Reviewing Policy:**

Affirmed.

**Final Justification:**

Most of my problems have been solved. This paper is actually a bit more engineering. I think 4 points is a reasonable score.

**Key Questions For Authors:**

- Since the base model already shows artifacts due to rapid motion, does the implementation of the camera motion function in this method make these artifacts even more severe?
- Because the training involves LoRAs, how is the generalization performance across different cases after the LoRA is fixed?
- Have the authors considered moving away from LoRAs and instead training a more universal model where users input the corresponding trajectory and the model generates the corresponding camera motion?

**Limitations:**

yes

**Strengths And Weaknesses:**

Strengths:
- The paper demonstrates the feasibility of training using synthetic videos with minimal geometries and extremely short rendering times, providing a highly cost-effective approach to solving the scarcity of 3D camera annotation data.
- The generated videos can achieve complex camera movements that are difficult for conventional datasets to cover; if designed properly, this could theoretically cover the vast majority of camera motions.
- By using an appearance LoRA, temporal derivative consistency loss, and style-aligned prompts, the model learns complex motions while avoiding being "led astray" by the low-quality appearance of the synthetic data.

Weaknesses:
- The core "dual-adaptation" mode (appearance LoRA + motion module) in this paper is functionally almost identical to AnimateDiff (Guo et al., 2023), which makes me feel that the technical innovation in this paper is somewhat insufficient.
- Compared to baseline models like CogVideoX or AC3D, the proposed method seems to achieve camera motion control at the cost of realism and detail (as seen in the FVD decline and results shown in the demo), resulting in a degradation of visual fidelity. I wonder if this is related to the single or simple style of the videos rendered by Unity, which may have damaged the prior of the original model.

---

> ### Author Rebuttal · Authors · 2026-03-31
>
> We thank reviewer hroA for the thoughtful comments.
>
> **Q1** The dual-adaptation is functionally almost identical to AnimateDiff.
>
> **A1** Thank you. We would like to clarify that our main contribution is less about dual adaptation itself, but more about an effective and inexpensive pipeline for learning unconventional camera motions from low-quality synthetic videos despite the large synthetic-to-real gap (as stated in L110-117).
>
> AnimateDiff uses a removable domain adapter to reduce discrepancies between video frames and high-quality images, such as motion blurs. It is not designed for the much larger gap between synthetic training videos and realistic videos. In fact, Table 4 and Figure 5 show that dual adaptation alone does not produce realistic results.
>
> Our contribution is to identify and validate a pipeline that makes this transfer work, including training data design, dual adaptation, style-aligned prompts, and the motion-consistency loss. We do not claim any of them as individually novel. Our contribution is to show that this synthetic-to-real transfer problem, despite its large appearance gap, can be effectively addressed within a practical pipeline.
>
> **Q2** VividCam seems to achieve camera motion control at the cost of realism. I wonder if this is related to single style of videos rendered by Unity.
>
> **A2** Thank you for the insightful comment. Indeed, potential degradation in realism is one of the main risks when disentangling camera motion, which is why we carefully examine FVD under different settings in Table 2. Fortunately, we find that in many cases the FVD degradation is very small, especially for simple and composed motions. In addition, prior work[1] has pointed out that automatic metrics such as FVD do not always correlate well with human. Therefore, we further conduct a human study in Table 3, which shows that our method does not damage the realism of the original model.
>
> As you suggested, we are also interested in how the style of Unity-rendered videos affects realism. We perform an additional experiment in which we vary the diversity of objects/textures in the rendered videos, from 2 geometries to 4 geometries (our current setting) to 10 geometries. We observe that the realism score improves slightly as the object diversity increases, as shown in the table below. This suggests that, with more human effort devoted to creating synthetic training videos with richer and more diverse visual styles, the realism of the generated videos could be further improved.
>
> [1]Movie Gen: A Cast of Media Foundation Models
> | |TransErr|RotErr|FVD|
> |-|-|-|-|
> |VividCAM(4-category objects)|0.4011|0.5013|1866.40|
> |2-category objects|0.4027|0.5135|2031.91|
> |10-category objects|0.4122|0.5078|1845.28|
>
> **Q3** Does the implementation of the camera motion in this method make artifacts more severe?
>
> **A3** Thank you. To examine whether VividCam makes rapid-motion artifacts more severe, we first note, as discussed in Q2, that the human study in Table 3 shows that our method does not reduce the realism of the original model. In addition, following prior work [1], we report visual quality scores to further assess the effect of VividCam on video quality. The results indicate that VividCam does not introduce significant additional artifacts compared with their respective base models, as reflected by their comparable visual quality scores.
>
> [1]VideoScore: Building Automatic Metrics to Simulate Fine-grained Human Feedback for Video Generation
> | |Visual Quality Score|
> |-|-|
> |VividCAM-Cog|2.813|
> |CogVideoX(base)|2.828|
> |VividCam-AC3D|2.865|
> |AC3D(base)|2.841|
>
> **Q4:** How is the generalization performance across different cases after the LoRA is fixed?
>
> **A4:** Thanks. We would like to clarify that a single trained LoRA is designed to cover multiple camera motions. Once trained, the model, with the camera LoRA in VividCam-CogVideo or the camera encoder in VividCam-AC3D, can generate different camera motions based on the input instructions or trajectories. Our current Tables 2 and 3 already evaluate how one module generalizes across different actions.
>
> **Q5** How about moving away from LoRAs and instead training a more universal model?
>
> **A5** Thank you. We agree that a more universal video model that directly takes trajectory or camera-motion text as input could potentially achieve stronger performance. However, we would like to first clarify that, even in this setting, the Appearance LoRA would still be needed to disentangle the synthetic appearance introduced by the training videos. In addition, VividCam is designed as a low-cost training framework, with minimal effort required for data preparation and minimal training cost by leveraging existing video models. Training such a foundation model from scratch would be beyond our computational budget and would deviate from our low-cost design principle. As such, we plan to explore the performance of universal models and compare their efficiency with VividCam in future work.

---

> > ### Author Rebuttal · Reviewer_hroA · 2026-04-03
> >
> > My problem has basically been resolved, so I keep my score.

---

> > > ### Author Response · Authors · 2026-04-07
> > >
> > > Thank you again for your thoughtful suggestions. We believe the paper has become stronger as a result of your comments and the corresponding revisions. Meanwhile, we would greatly appreciate it if the reviewer could consider raising the score in light of these clarifications and revisions.

---

### Official Review · Reviewer_aP3x · 2026-02-17

**Soundness:** 3
**Presentation:** 3
**Significance:** 2
**Originality:** 3
**Overall Recommendation:** 3
**Confidence:** 4

**Summary:**

This paper presents VIVIDCAM, a framework for enabling video diffusion models to learn unconventional camera motions using synthetic low-poly videos rendered in Unity. The method adopts a dual-adaptation pipeline: an appearance LoRA is first trained on static synthetic videos, followed by a camera motion module trained on motion sequences. Experiments on both text-driven and trajectory-based camera control demonstrate improvements over several existing approaches.

**Compliance With Llm Reviewing Policy:**

Affirmed.

**Key Questions For Authors:**

1. How is translation and rotation error computed for text-to-video models (CogVideo Xand VividCam-Cog)?
2. Have you evaluated generalization to camera motions that were not explicitly present during training?
3. (This is only out of curiosity rather than limitation) Unity is a powerful rendering engine, yet the paper focuses on low-poly scenes. Why not use Blender, which is more commonly adopted in the research community and may offer a simpler pipeline for synthetic data generation? Can the author explain the choose of Unity among Unity, Unreal Engine, and Blender?

**Limitations:**

yes

**Strengths And Weaknesses:**

## Strengths

**Interesting problem setting.**
The paper addresses the challenging task of learning unconventional camera motions, which is an important and underexplored direction for controllable video generation.

**Thorough ablation studies.**
The authors provide ablations on appearance LoRA, style-aligned prompts, and the temporal consistency loss, which help clarify the contributions of individual components.

---

## Weaknesses

The paper argues that low-poly synthetic data benefits camera-controlled video generation. However, prior work (e.g., *SynCamMaster*, *ReCamMaster*, *PanoWorld-X*) demonstrates strong performance using high-fidelity Unreal Engine renderings. Given that Unity is also capable of high-quality rendering, it is unclear why the pipeline relies on intentionally low-quality assets followed by adaptation, rather than directly generating realistic synthetic data.

The overall training pipeline closely resembles prior dual-adaptation frameworks (e.g., AnimateDiff-style appearance adaptation followed by motion learning). The additional components, such as style-aligned prompts and the temporal derivative loss, appear incremental. Although the paper claims a “novel framework,” much of the contribution seems to lie in engineering design rather than new training principles.

Only a brief discussion of failure modes is included in the appendix. More qualitative analysis and visualization of failure cases would strengthen the paper and clarify the method’s limitations.

## Minor Issues

- Reference formatting is inconsistent (mix of full conference names and abbreviations). Please ensure uniform formatting.
- The baseline selection appears somewhat limited; more recent methods such as *DualCamCtrl* could be included for a stronger comparison.

---

> ### Author Rebuttal · Authors · 2026-03-31
>
> We thank reviewer aP3x for the thoughtful comments.
>
> Q1: Why VividCam generates low-quality assets, not directly generate realistic data?
>
> A1: Thank you. High-fidelity rendered videos can be effective training data, but producing them at scale requires substantial manual effort, making the pipeline costly. In contrast, VividCam is designed for scalability: each video takes about 5 seconds to synthesize without manual annotation, enabling large-scale supervision.
>
> Nevertheless, to examine the effect of asset quality, we compare lower and higher quality assets, where higher quality settings include more diverse assets and textures. The results show a slight impact on video quality (FVD). Thus, VividCam prioritizes scalability and controllability, while high quality assets can further improve visual quality at additional cost.
> | |TransErr|RotErr|FVD|
> |-|-|-|-|
> |VividCAM(4-category objects)|0.4011|0.5013|1866.40|
> |2-category objects|0.4027|0.5135|2031.91|
> |10-category objects|0.4122|0.5078|1845.28|
>
> Q2: The training pipeline resembles AnimateDiff. Additional components appear incremental.
>
> A2: Thanks. Our main contribution is not dual adaptation itself, but a practical and inexpensive pipeline for learning unconventional camera motions from low-quality synthetic videos despite the large synthetic-to-real gap (L110-117). AnimateDiff’s adapter is designed to reduce discrepancies between video frames and high-quality images, not the much larger gap between synthetic training videos and realistic videos. Table 4 and Figure 5 show that AnimateDiff alone does not produce realistic results. Our contribution is to show that this transfer problem can be effectively addressed within a practical training pipeline.
>
> Q3: More analysis and visualization of failure cases.
>
> A3: Thanks. Following your suggestion, we reorganize the failure-case discussion and quantify the effects below. We observe three main failure modes:
> * Surroundings: less descriptive prompts for surroundings can reduce realism.
> * Objects: scenes with too many dynamic objects can exceed the base model’s capacity, and the resulting incoherence is inherited by VividCam.
> * Actions: overly long camera motions are not always fully captured, likely because the base model only generates 49-frame videos.
>
> We also provide examples on the anonymous page (shown in **Section 6. Failure Case Examples** in the link provided in main paper, Figure 4 caption). We will clarify them in the revision.
>
> Q4: Performance of recent methods (DualCamCtrl)?
>
> A4: Thanks. We provide comparisons with DualCamCtrl (examples in **Section 7. Comparison with DualCamCtrl** in the link). Qualitatively, DualCamCtrl handles some simple motions, such as switch object, but struggles with more precise motions such as searching for and focusing on an object. VividCam is more consistent across diverse motions. In addition, DualCamCtrl requires depth images as input, making it less practical than VividCam.
> | |TransErr|RotErr|FVD|
> |-|-|-|-|
> |VividCAM-Cog|0.4011|0.5013|1866.40|
> |(Baseline)DualCamCtrl|0.4372|0.7134|1792.15|
>
> Q5: Reference Format.
>
> A5: Thanks. We will fix citations to ensure format consistency.
>
> Q6: How is translation and rotation error computed for text-to-video models?
>
> A6: For text-to-video models, these errors are computed by comparing the camera trajectory inferred from the generated video with the ground-truth trajectory. In paper, we acknowledge that this metric is somewhat unfair to both VividCam-Cog (ours) and CogVideoX, since neither takes the ground-truth trajectory as input. We therefore conduct a human study on camera-action correctness and video realism, as discussed in L363-L369.
>
> Q7: Generalization to unseen camera motions?
>
> A7: Thanks. Our goal is not zero-shot generalization to entirely unseen motion categories, but efficient incorporation of new motions by defining new 3D trajectories and synthesizing corresponding training videos. In our pipeline, a new motion can be introduced simply by specifying a 3D trajectory in Unity, and the associated training videos can be generated in seconds, after which the model can be trained on the new motion efficiently. In our experiments, we do not observe reliable generalization to entirely unseen motions, likely because different camera motions correspond to fundamentally different 3D transformations.
>
> Q8: Choice among Unity, Unreal Engine, and Blender?
>
> A8: Thank you. Our choice of Unity is mainly motivated by efficiency and scalability, not rendering realism. We agree that Unreal Engine and Blender are powerful for high-quality rendering, but producing large-scale trajectory-controlled data with them also requires substantial manual effort in scene creation, assets, lighting, rendering setup, and trajectory design. In contrast, Unity is generally easier to use for rapid prototyping and scripting-driven pipelines. We agree that exploring more efficient generation pipelines with different engines in the future would be valuable.

---

> > ### Author Rebuttal · Reviewer_aP3x · 2026-04-03
> >
> > I have read through authors' rebuttal. However, my concerns are not fully resolved and I do not agree with some of the authors' claims. Specifically,
> >
> > - The authors claimed that "High-fidelity rendered videos can be effective training data, but producing them at scale requires substantial manual effort, making the pipeline costly". However, many recent papers synthesized realistic video data with Unreal Engine (e.g., ReCamMaster, Panoworld-X, airSim360), such pipeline can be easily scalable with more 3D assets, which are readily available in large scale online, e.g., on Fab, a 3D asset marketplace selling digital asset compatible with both Unreal Engine and Unity. Given that authors' additional experiments clearly showed that more realistic data can bring stronger result, it looks strange that authors intentionally bring train-test distribution gap to the pipeline, instead of narrowing it.
> >
> > - The authors claimed that "We agree that Unreal Engine and Blender are powerful for high-quality rendering, but producing large-scale trajectory-controlled data with them also requires substantial manual effort in scene creation, assets, lighting, rendering setup." However, Blender supports Python scripting, and Unreal Engine supports both Python and C++ scripting. For Unreal Engine, as I have mentioned, we can directly purchase render-ready 3D assets online and we do not need to manually set lighting and assets.
> >
> > I acknowledge that my other questions are addressed adequately. However, given the concerns above, I decide to maintain my original rating.

---

> > > ### Author Response · Authors · 2026-04-07
> > >
> > > We thank reviewer aP3x for the additional comments.
> > >
> > > We understand the concern regarding the unnecessary domain gap. This is essentially a cost-quality trade-off. Here is our new response to this:
> > > * __Even if render-ready assets can be purchased online, the pipeline remains costly.__ For example, on Fab, 3D assets marked as realistic typically range from __\$15 to 100__ or more. Moreover, such pipeline requires diverse asset categories to reduce the risk of overfitting to a narrow set of objects (e.g., cars, furniture, animals). Suppose the pipeline includes 20 asset categories, with 5 variations per category (e.g., different types of cars). The total cost would then be approximately __\$1,500 to 10,000__, or even higher. On the other hand, while some free assets are available, many are of limited quality, and human effort is still required to identify and filter usable ones.
> > > * Our results show that the improvement in asset quality only marginally improves the performance (FVD: 1866.40 -> 1845.28). So in many applications where minimizing cost is a priority, our current configuration offers a desirable tradeoff.
> > > * Nevertheless, to show the advantage of our method across a broad range of realism, we decided to follow your suggestion and __add one experiment__, where we adopt the Unity assets you mentioned with __two asset settings__ with different cost levels:
> > >     1. 10 assets, average cost \$15.
> > >     2. 40 assets, average cost \$20.
> > >
> > > We have completed the first setting and show the results below. The second row reports the result of directly using these training videos without the VividCam pipeline, while the third row reports the result of using these higher-quality assets within the VividCam pipeline. The results indicate that even with more realistic assets, directly using the training videos without the VividCam pipeline still yields limited visual quality. In addition, when these more realistic assets are incorporated into the VividCam pipeline, the visual quality further improves over the original geometry asset setting.
> > >
> > > | Setting                                | TransErr | RotErr | FVD     |
> > > |----------------------------------------|----------|--------|---------|
> > > | VividCAM (original geometry assets)    | 0.4011   | 0.5013 | 1866.40 |
> > > | 10 realistic assets, no VividCam       | 0.3822   | 0.5761 | 2125.10 |
> > > | 10 realistic assets, VividCam pipeline | 0.4198   | 0.5404 | 1769.27 |
> > >
> > > We are currently running the second setting and will include both results in the revised paper. We hope our experiment could address your concern on domain gaps, and that you could take this into consideration for evaluation.

---

### Official Review · Reviewer_wdie · 2026-03-09

**Soundness:** 3
**Presentation:** 2
**Significance:** 4
**Originality:** 3
**Overall Recommendation:** 4
**Confidence:** 3

**Summary:**

This paper starts from the observation that current video generation models struggle to handle complex and unconventional camera motions. To address this limitation, the authors propose a synthetic-data-based training framework that enables precise camera control without degrading visual quality. The method demonstrates promising generalization across different base models and various types of camera trajectories. Overall, this manuscript addresses a central challenge in controllable video generation, namely the disentanglement of aesthetic appearance and camera motion, and provides a potentially effective pathway toward separating these two factors.

**Compliance With Llm Reviewing Policy:**

Affirmed.

**Key Questions For Authors:**

1. Table 1 summarizes the range of camera trajectories considered. For motion types involving semantic control (e.g., switching focus between objects), can these be generated using the trajectory-based model? If so, how is the corresponding trajectory defined in advance for such semantically guided videos?
2. Since the Appearance LoRA is discarded during inference to preserve visual quality, could the authors provide qualitative comparisons (or aesthetic evaluation results) with and without discarding the Appearance LoRA? This would more directly demonstrate the effectiveness of the proposed appearance-motion disentanglement strategy.
3. The paper introduces $L_{flow}$​ to enforce temporal derivative consistency. Have the authors experimented with larger values of the balancing coefficient $\lambda$, such that the model relies predominantly on flow-like signals for learning? How would this affect performance?

**Limitations:**

yes

**Strengths And Weaknesses:**

Strengths
- The paper proposes a feasible solution to a significant and challenging problem: how to leverage synthetic training data without sacrificing the visual quality of generated videos.
- The authors provide comprehensive ablation studies and intuitive qualitative results, clearly demonstrating the precision of ViVidCAM in handling complex camera motions and validating the effectiveness of each component in the framework.

Weaknesses
- The presentation could be improved for clarity and logical coherence.
    - **Figures.** Some figures are not sufficiently self-explanatory. For instance, if my understanding is correct, the Appearance LoRA is discarded at inference time. However, this is not clearly indicated in Figure 3, which may make the pipeline less intuitive. In addition, the figure does not explicitly distinguish between the text-based and trajectory-based variants of the model.
    - **Formulation.** Certain equations are presented in a potentially confusing manner. If I understand correctly, the two $L_2$​ terms around L255 correspond to the text-based and trajectory-based settings, respectively. However, at L242, the notation uses $L_1$​ and $L_2$ to denote different losses, and at L271 only $L_2$ is referenced. This inconsistency in notation may cause confusion for readers.
    - **Typographical issue.** There appears to be a spelling error at L217.
- The evaluation of video quality could be more comprehensive. The paper primarily relies on FVD to measure visual quality, and the FVD performance is not consistently better than the baselines. Incorporating additional metrics, such as aesthetic quality scores or perceptual-based evaluation metrics, could provide a more holistic assessment of overall video quality.

---

> ### Author Rebuttal · Authors · 2026-03-31
>
> We thank reviewer wdie for the thoughtful comments.
>
> **Q1** The presentation (figures, formulations, typo) could be improved.
>
> **A1** Thank you for the detailed suggestions. We will address them carefully in the revision, point by point:
>
> * **Figures.** We agree that Figure 3 can be clearer, and we redraw it to explicitly indicate that the A-LoRA is discarded at inference time and to better distinguish the text-based and trajectory-based variants. We show the revised figure in our anonymous page (shown in **Section 3. Figure Presentation** in the link provided in main paper, Figure 4 caption).
> * **Formulation.** We also agree that the notation in the formulation is confusing. We originally intended to use $L_1$ to denote the appearance loss and $L_2$ to denote camera motion loss. This is true for both Line 242 and Line 255. It is just that the typo in Line 244 adds to the confusion. We will correct the formula. Because of the reply parser limitations, we cannot include the revised equations directly in the response. If helpful, we can provide a screenshot in the discussion phase.
> * **Typo.** We will also fix the typo “wepresent” at L217.
>
> **Q2** More metrics for video evaluation? FVD not consistently better?
>
> **A2** Thanks. We include two additional metrics to evaluate videos from different perspectives. Following prior work[1], we report visual quality scores and dynamic degree scores. The results are shown in the table below. We find that VividCam achieves higher dynamic-degree scores than the corresponding baselines, indicating its ability to generate more complex and expressive camera motions. Meanwhile, VividCam achieves visual quality scores comparable to baselines, suggesting that it does not reduce the aesthetic quality.
>
> Regarding the FVD results in Table 2, we would like to clarify that our method is not designed to improve visual quality. Our focus is purely on improving action correctness without compromising the visual quality. We include FVD because we want to verify that introducing our method does not degrade quality. We would already consider it a success if our model can maintain the same level of visual quality. The comparable FVD results show that, although VividCam is designed for better action correctness rather than better visual quality, with our method design, it does not cause a loss in perceptual quality.
>
> [1]VideoScore: Building Automatic Metrics to Simulate Fine-grained Human Feedback for Video Generation
> | |Dynamic Degree Score|Visual Quality Score|
> |-|-|-|
> |VividCAM-Cog|2.781|2.813|
> |-CogVideoX(base)|2.422|2.828|
> |VividCam-AC3D|3.297|2.865|
> |-AC3D(base)|3.187|2.841|
> |CameraCtrl|2.924|3.078|
>
> **Q3** How to generate videos with semantic control? How is the trajectory defined in advance for such videos?
>
> **A3** Thanks. We find that VividCam has the capability to infer object placement according to the provided trajectory and content description. Our setup is as follows: we provide a camera trajectory showing that the camera moves left and right, without specifying the object positions. In the content description, we further specify the objects, for example, two birds. We find the generated video correctly arranges the positions of the birds and the timestamps at which they appear. An example can be found on our anonymous page (**Section 4. Videos with semantic control**). We attribute this to the capability of the video generation model and the synthetic training data, which associate similar camera motions with plausible object layouts and movement patterns.
>
> **Q4** Performance when discard the Appearance LoRA?
>
> **A4** Thanks. We provide qualitative comparisons in **Section 5. Results without Discarding Appearance LoRA** on the anonymous page, along with aesthetic evaluation results for inference with and without discarding A-LoRA.
> The results show that keeping A-LoRA during inference causes substantial visual quality degradation and introduces artifacts, which supports the effectiveness of the proposed appearance-motion disentanglement strategy.
> | |FVD|Visual Quality Score|
> |-|-|-|
> |VividCAM-Cog|1866.40|2.813|
> |No A-LoRA|2097.61|2.280|
> |VividCam-AC3D|1721.45|2.865|
> |No A-LoRA|2156.02|2.173|
>
> **Q5** How does $\lambda$ in $L_{flow}$ influence the performance?
>
> **A5** Thanks. In this experiment, we vary $\lambda$ to study its effect on both camera motion accuracy and video quality. As shown in the table, with small $\lambda$, VividCam is less able to learn accurate camera motions; with large $\lambda$, the visual quality of the videos degrades, indicated by higher FVD. We believe this is because smaller $\lambda$ weakens the temporal loss, while larger $\lambda$ causes the temporal consistency objective to dominate the original diffusion loss, making it harder for the model to learn how to denoise and generate clear videos.
> | |TransErr|RotErr|FVD|
> |-|-|-|-|
> |VividCAM,$\lambda$=0.3|0.4011|0.5013|1866.40|
> |$\lambda$=0.03|0.5859|0.7156|1808.94|
> |$\lambda$=3|0.4266|0.4923|2124.11|

---

> > ### Author Rebuttal · Reviewer_wdie · 2026-04-08
> >
> > The responses have resolved most of my questions. I maintain my score of 4 (weak accept).

---

### Official Review · Reviewer_R9wu · 2026-03-11

**Soundness:** 3
**Presentation:** 3
**Significance:** 3
**Originality:** 3
**Overall Recommendation:** 4
**Confidence:** 3

**Summary:**

### VIVIDCAM:  Learning Unconventional Camera Motions from Virtual Synthetic Videos

VIVIDCAM is a training paradigm that enables video diffusion models to learn complex, unconventional camera motions—such as dolly zooms and explosive shakes—using simple, low-poly synthetic videos. By leveraging synthetic data, this approach eliminates the dependency on large-scale, realistically annotated datasets, which are often scarce for artistic or avant-garde camera movements.

---

### Methodology

* Synthetic Data Generation: Using the Unity engine, the authors render basic 3D scenes (background, floor, and simple objects) to create two sets of training data: videos with the target camera motions and static videos without motion.

* Appearance Adaptation: The model first trains an Appearance LoRA on the static videos ($\mathcal{X}_{a}$) to "absorb" the low-poly synthetic style.

* Camera Control Learning: A separate camera module, either a Camera LoRA for text-based prompts or a Trajectory Encoder for 3D paths, is trained on the motion-heavy videos ($\mathcal{X}_{c}$) while the Appearance LoRA remains frozen.

* Temporal Derivative Consistency Loss: This loss function aligns the frame-to-frame pixel differences of generated videos with the ground-truth synthetic motion to ensure physical accuracy.

* Inference: At generation time, the Appearance LoRA is discarded. This allows the model to apply the learned motion to its original realistic weights, producing high-quality video with unconventional movement.

---

### Contributions

* Novel Training Paradigm: Proves that diffusion models can learn sophisticated cinematic intent from "surprisingly simple" 3D geometry rather than photorealistic footage.

* Disentanglement Framework: Introduces a robust method to separate camera motion from undesirable visual artifacts, mitigating domain shift between synthetic and real data.

* Complex Motion Library: Enables a wide range of "meaning-driven" motions (e.g., "seeking" an object and locking focus) and "stylized" effects (e.g., 180° rotations).

---

### Experiments

The authors evaluated VIVIDCAM using CogVideoX-5B (text-based) and AC3D (trajectory-based) as base models.

* Quantitative Accuracy: VIVIDCAM achieved significantly lower translation (TransErr) and rotation (RotErr) errors compared to baselines like CameraCtrl across simple, composed, and complex motions.

* Visual Fidelity: According to FVD metrics and human realism scores, the model maintained high visual quality comparable to vanilla realistic models.

* Human Study: In a study with 88 participants, VIVIDCAM consistently outperformed state-of-the-art methods in Action Correctness, particularly for complex "unseen" motions.

---

### Conclusion

VIVIDCAM offers a cost-efficient solution for precise camera control in video generation. By demonstrating that "garbage" synthetic data can teach "high-end" cinematic skills, it opens a new path for personalized video creation.

**Compliance With Llm Reviewing Policy:**

Affirmed.

**Final Justification:**

The authors have adequately addressed my concerns. I maintain my score of 4 (weak accept).

**Key Questions For Authors:**

* Text-Trajectory Alignment: How well does the model maintain the relationship between a specific text instruction (e.g., "searching for a bird") and a provided 3D trajectory?

* Multi-Entity Motion: While VIVIDCAM handles camera motion and simple dynamic objects (like fire or children), how does it perform when the camera is doing an unconventional move simultaneously with complex objects?

* Pixel-level Alignment: With the given camera parameters (Trajectory Alignment), could the model achieve pixel-level alignment?

**Limitations:**

* Please see the Weakness and Key Questions part.

**Strengths And Weaknesses:**

### Strengths

* Cost-Efficient Data Strategy: The method significantly reduces the human effort and expertise required for video generation by using simple, low-poly 3D scenes instead of labor-intensive, high-quality realistic videos.

* Complex Motion: VIVIDCAM demonstrates a unique ability to handle "unconventional" and "meaning-driven" camera motions, such as seeking an object and locking focus, which are often unavailable in standard datasets.

* Effective Domain Gap Mitigation: Through its dual-adaptation training and style-aligned prompts, the framework successfully isolates camera motion from synthetic "virtual" artifacts, maintaining high realism in final outputs.

* Versatile Control Paradigms: The framework is designed to work across both text-based and trajectory-based camera control, making it adaptable to different user needs and existing model architectures.

* Strong Quantitative and Qualitative Performance: Experimental results show that VIVIDCAM outperforms state-of-the-art baselines in camera precision (TransErr/RotErr) while maintaining visual quality comparable to models trained on real data.

---

### Weaknesses

* Limited Object Motion: While the paper excels at camera movement, it focuses primarily on static or simple moving scenes; it does not extensively explore complex, multi-entity interactions where objects move independently in highly dynamic ways.

* Real-World Data Necessity: The paper suggests that synthetic data is "sufficient" for training. However, Table 5 shows that a mixture of real and synthetic data still yields slightly different results.

---

> ### Author Rebuttal · Authors · 2026-03-31
>
> We thank reviewer R9wu for the thoughtful comments.
>
> **Q1** Limited Object Motion: In addition to static or simple moving scenes, how about performance where objects move independently in highly dynamic ways?
>
> **A1** Thank you for the comment. We agree that highly dynamic videos are an important direction in video generation. However, the base model, CogVideoX-5B, does not reliably support highly dynamic scenes. We show this in our anonymous page (shown in **Section 1. Camera Motions in High Dynamic Scenes** in the link provided in main paper, Figure 4 caption), videos generated by the base model often exhibit strong artifacts and ghosting under such settings, and VividCam inherits them. Therefore, in the main paper, we primarily focus on dynamic scenes with single-entity interactions that the base model can handle more reliably. Moreover, we evaluate realism scores [1] in both single-object and multi-object scenarios, and the results show that our method does not reduce the dynamic capability of the base model. Therefore, we conclude that this limitation mainly comes from the base model itself, rather than from the proposed camera-control framework.
>
> [1]VideoScore: Building Automatic Metrics to Simulate Fine-grained Human Feedback for Video Generation
> | Setting      | Single Dynamic Scene | Multiple Dynamic Scene |   |   |
> |--------------|----------------------|------------------------|---|---|
> | CogVideoX    | 2.704                | 2.434                  |   |   |
> | VividCAM-Cog | 2.686                | 2.547                  |   |   |
>
> **Q2** Real-World Data Necessity: The paper suggests that synthetic data is "sufficient" for training. However, Table 5 shows that a mixture of real and synthetic data still yields slightly different results.
>
> **A2** Thank you for the comment. We would like to clarify that the performance differences associated with including real data are very small, and incorporating real data does not necessarily improve performance. As shown in Table 5, the differences are within 0.03, and including real data even leads to worse performance in terms of action accuracy. We believe this is because the real data from RealEstate10K do not exhibit camera motions as diverse as those in our synthetic data. Given the difficulty of further scaling up real data collection, we believe that using synthetic data alone is a more manageable and effective solution. That said, we agree that the word “sufficient” may overstate the conclusion of this experiment. We will revise the paragraph to reflect this nuance and avoid using “sufficient” in the description.
>
> **Q3** Text-Trajectory Alignment: How well does the model maintain the relationship between a specific text instruction (e.g., "searching for a bird") and a provided 3D trajectory?
>
> **A3** Thank you for the question. We would like to clarify that our design intentionally supports camera control through either text input (VividCam-CogVideo) or 3D trajectory input (VividCam-AC3D), but not both simultaneously, to avoid conflicting control signals. Therefore, in VividCam-AC3D, when a 3D trajectory is provided, an additional text instruction related to camera motion does not alter the resulting motion. To verify this, we show videos generated by VividCam-AC3D with an irrelevant text instruction “rotates 180 degrees”, shown in **Section 2. Text-Trajectory Alignment** on our anonymous page. The results indicate that the text instruction does not interfere with the generated motion in this case. We agree that jointly using text and trajectory to control camera motion is an interesting direction for future work, and we will explore how to address the consistency issue.
>
> **Q4** While VIVIDCAM handles camera motion and simple dynamic objects, how does it perform when the camera is doing an unconventional move simultaneously with complex objects?
>
> **A4** Thank you for the question. Please refer to **Q1**.
>
> **Q5** With the given camera parameters, could the model achieve pixel-level alignment?
>
> **A5** Thank you for the question. We investigated this point, but we are not entirely sure what you meant by “pixel-level alignment.” We tried to interpret your questions in two different ways.
> * If you meant the pixel-wise differences between the generated video and a ground-truth video, we would like to clarify that VividCam uses either a trajectory or a text instruction as input, and there is no paired ground-truth visual content or reference frame for direct pixel-level comparison.
> * If you meant camera-control precision, namely the comparison between the ground-truth trajectory and an inferred camera trajectory estimated from the generated video, this is already captured by the TransErr and RotErr metrics in Table 2. These results indicate that VividCam achieves more precise camera motion than the baselines.
>
> Of course, we may still misunderstand your question. We would be happy to further clarify this point during the discussion phase.

---

> > ### Author Rebuttal · Reviewer_R9wu · 2026-04-02
> >
> > The authors have adequately addressed my concerns. The dynamic scene limitation (W1) is convincingly attributed to the base model rather than the proposed framework, supported by VideoScore metrics. The revision of the "sufficient" claim regarding synthetic data (W2) is appreciated. The clarification that text and trajectory control are intentionally separate modes (Q3) is reasonable. Regarding "pixel-level alignment," my intention was interpretation 2. Table 2 shows that the model outperforms most baselines, but achieving precise pixel-level camera alignment remains an open direction for future work. I maintain my score of 4 (weak accept).

---

> > > ### Author Response · Authors · 2026-04-07
> > >
> > > Thank you again for your thoughtful suggestions. We believe the paper has become stronger as a result of your comments and the corresponding revisions. Meanwhile, we would greatly appreciate it if the reviewer could consider raising the score in light of these clarifications and revisions.

---

### Decision · Program_Chairs · 2026-04-30

**Decision:**

Accept (regular)

**Comment:**

This paper was reviewed by 4 experts in the field. After discussion, the reviewers still hold a mixed review to this work. The rating is 4(weak accept), 4(weak accept), 4(weak accept), 3(weak reject).

In general, reviewers agrees that this work enables complex and unconventional camera control, which is not covered by previous work. It also successfully bridges the gap between synthetic and real-world data through effective disentanglement, which is one of fundamental challenges of video generation training.

Still, reviewers raised several concerns to this work. The concern includes 1) limited technical novelty given its similarity to existing dual-adaptation pipelines, 2) reduced effectiveness in generating videos with highly dynamic object motion. For the technical limitation, the area chair feels that although the technical is similar to AnimatedDiff, this work is aiming a different task. Quality concern is valid, but not fatal.

Regardless of this limitations, the decision of this work is to Accept. Still, we strongly recommend the authors carefully read all reviewers’ final feedback and revise the manuscript as suggested in the final camera-ready version if being accept.

Besides, the area chair may also suggest to try on latest video generation model, like WAN series, to increase the impact of this work.